# Simple and Effective Specialized Representations for Fair Classifiers

**Alberto Sinigaglia**[*]
Human Inspired Technology Research Center
University of Padua
alberto.sinigaglia@phd.unipd.it

**Davide Sartor**[*]
Department of Information Engineering
University of Padua
davide.sartor.4@phd.unipd.it

**Marina Ceccon**
Department of Information Engineering
University of Padua
marina.ceccon@phd.unipd.it

**Gian Antonio Susto**
Department of Information Engineering
University of Padua
gianantonio.susto@unipd.it

## Abstract

Fair classification is a critical challenge that has gained increasing importance due to international regulations and its growing use in high-stakes decision-making settings. Existing methods often rely on adversarial learning or distribution matching across sensitive groups; however, adversarial learning can be unstable, and distribution matching can be computationally intensive. To address these limitations, we propose a novel approach based on the characteristic function distance. Our method ensures that the learned representation contains minimal sensitive information while maintaining high effectiveness for downstream tasks. By utilizing characteristic functions, we achieve a more stable and efficient solution compared to traditional methods. Additionally, we introduce a simple relaxation of the objective function that guarantees fairness in common classification models with no performance degradation. Experimental results on benchmark datasets demonstrate that our approach consistently matches or achieves better fairness and predictive accuracy than existing methods. Moreover, our method maintains robustness and computational efficiency, making it a practical solution for real-world applications.

## 1 Introduction

Algorithmic fairness has become a central concern in deploying automated decision-making systems, especially in high-stakes domains like hiring, lending, criminal justice, and healthcare [4, 49, 24, 21]. The growing reliance on these systems has raised concerns about their potential to reinforce or amplify societal biases [3, 15, 48, 6]. In response, a large body of research has focused on detecting, analyzing, and mitigating bias throughout the algorithmic pipeline [44, 55, 36, 9].

Numerous fairness definitions and metrics have been proposed, reflecting a wide range of normative and technical perspectives [18, 27, 16]. A common criterion is statistical independence, which requires that model predictions be independent of sensitive attributes like ethnicity or gender [42, 3]. A simple strategy for enforcing this criterion is to exclude the sensitive attribute from the model's input features, an approach commonly referred to as *Fairness through Unawareness* [10]. However, this method is often ineffective in practice, as sensitive information may still be indirectly captured through other correlated features, so-called proxy variables [39]. Removing all proxies typically results in significant degradation of performance, as task-relevant information is also discarded [10].

---

[*]These authors contributed equally.

39th Conference on Neural Information Processing Systems (NeurIPS 2025).

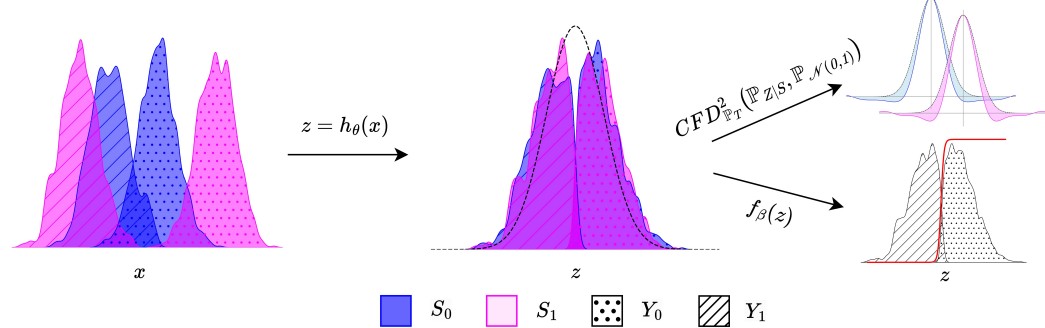

Figure 1: Overview of the proposed approach. Each sample $X$ is associated with a sensitive attribute $S$ and a target label $Y$. The conditional distribution $\mathbb{P}_{X|S}$ is mapped to a new distribution $\mathbb{P}_{Z|S}$, which is encouraged to resemble a Gaussian Distribution via the Characteristic Function Distance (CFD). The encoded representation $Z$ minimizes $\Delta(\mathbb{P}_{Z|S_0}, \mathbb{P}_{Z|S_1})$ while retaining task-relevant information.

To better balance fairness and predictive accuracy, many approaches aim to learn fair representations by projecting input data into a latent space that obscures sensitive information while preserving task-relevant structure [55, 35, 37, 2]. Once optimized, these representations can replace the original data during both training and testing, enabling fair decision-making in scenarios where sensitive attributes must not be explicitly used, such as determining loan approvals or hiring decisions. These approaches typically fall into two categories, which we refer to as *specialized* and *general*. Specialized representations are tailored to a specific task, filtering out sensitive and irrelevant features for improved performance. General representations, in contrast, are task-agnostic and aim for broad applicability across multiple tasks while being insensitive to protected attributes, but are harder to learn and may compromise predictive accuracy. Returning to the loan approval example, specialized representations learned in that context would be suitable only for predicting creditworthiness. If reused for another task, e.g. default risk or customer lifetime value, they might lack task-relevant information that was discarded during the fairness optimization. General representations, instead, preserve information that remains broadly useful across prediction tasks while still mitigating sensitivity to protected attributes.

A popular class of methods uses Variational Autoencoders (VAEs) to learn fair, specialized representations [32, 37]. In VAE-based approaches, fairness is optimized indirectly, either via a variational bound or Maximum Mean Discrepancy (MMD), which can misalign with true fairness goals and yield suboptimal results [2].

Other methods rely on Adversarial Networks that ensure specialized fair representations through a minimax game between a predictor and a fairness discriminator [19, 28]. Such approaches are often unstable and sensitive to hyperparameters, making the fairness-performance trade-off more challenging to manage [20]. Additionally, fairness guarantees are tied to the specific adversary used during training and may fail if a stronger or different adversary is applied later [37, 52, 26, 45].

Fair Normalizing Flows (FNF) [2] offer representations that are robust to any adversary, positioning itself as a state-of-the-art alternative to adversarial methods. Nonetheless, it has key limitations. It requires sensitive attribute information at inference, conflicting with privacy goals. It also trains a separate model for each sensitive group, which is computationally expensive, especially with multi-class attributes or large datasets. Crucially, while FNF aims to learn general representations, the inclusion of a classification loss can bias the model toward task-specific features. This may introduce group-specific information, potentially weakening fairness guarantees through representation bias.

There is an inherent trade-off between general and specialized representations: while general representations are broadly useful and transferable, achieving high predictive performance often requires some degree of specialization. Even models like FNF, which explicitly aim for generalization, ultimately rely on task-specific loss functions that guide representations toward specialization to optimize accuracy. This reliance suggests that specialization is not merely a byproduct but a necessary condition for strong predictive performance. Recognizing this, it becomes clear that embracing specialization is not only pragmatic but essential. Adversarial approaches naturally embody this perspective by tailoring representations to specific prediction tasks. However, they come with significant limitations, most notably, weaker guarantees around fairness and robustness [2, 37, 45]. This underscores the

need for alternative methods that, like adversarial techniques, pursue specialization, but do so through more transparent and controllable means, offering stronger fairness guarantees as a result.

To address these challenges, we propose a novel framework that embraces specialization in representations while rigorously enforcing fairness, without relying on adversarial training. In addition to stronger guarantees, our method is significantly simpler and more lightweight than existing approaches. This further justifies the aim to achieve specialized representations, as the advantages of transferability offered by general representations might not outweigh the benefit of higher prediction accuracy in settings where re-training for a different task is computationally lightweight. Our key contributions are summarized as follows:

- We introduce a new approach based on distribution matching via characteristic functions. This formulation avoids reliance on auxiliary components such as variational autoencoders, adversarial networks, or normalizing flows.

- We derive a simplified version of the proposed framework tailored to classification, which allows for formal guarantees on the sensitive information accessible to downstream classifiers.

- We conduct extensive experiments demonstrating that our method effectively removes sensitive information while matching SotA accuracy and delivering significantly fairer representations.

In summary, our approach offers a principled alternative for learning fair, specialized representations. It seamlessly combines simplicity, training stability, and robust fairness guarantees, all without relying on sensitive attributes during inference. This makes our method both practical and privacy-conscious, showing promising improvements over the current state-of-the-art [2].

## 2   Related Work

**Variational Autoencoders**   In the context of fairness through representation learning, prior work has explored the use of Variational Autoencoders (VAEs) to disentangle sensitive attributes from learned data representations [32, 37, 11, 31]. The encoder extracts task-relevant representations, while the decoder reconstructs the input. Though reconstruction may retain extra information, the optimization encourages representations tailored to prediction. A notable approach is the Variational Fair Autoencoder (VFAE) [32], which extends the standard VAE framework to enforce invariance in the latent space with respect to protected attributes. VFAE achieves this by introducing a penalty that explicitly encourages independence between the latent representation and the sensitive attributes. A related method was proposed by Moyer et al. [37], who combine ideas from VAEs and the Variational Information Bottleneck (VIB) to learn representations that are both informative and robust to variations in sensitive inputs.

**Adversarial Learning**   Another line of research focuses on fairness-aware learning through adversarial and information-theoretic methods to induce invariant representations [51, 41, 46]. In adversarial settings, a model such as an encoder or predictor is trained in opposition to an adversary whose goal is to recover sensitive attributes from the learned representation. This adversarial pressure encourages the model to generate latent features that do not reveal sensitive information, thereby promoting invariance with respect to protected variables [19, 22]. Xie et al. [51] proposed a general adversarial framework for learning representations that are invariant to arbitrary nuisance attributes. Their method formulates the learning process as a three-player minimax game involving an encoder, a task-specific predictor, and a discriminator that attempts to infer the nuisance attribute. Madras et al. [35] introduced Learning Adversarially Fair and Transferable Representations (LAFTR), which incorporates adversarial objectives aligned with specific fairness definitions, to learn representations that remain fair even when deployed by downstream classifiers without explicit fairness constraints. Roy and Bodetti [41] extended this adversarial paradigm by proposing MaxEnt-ARL, which maximizes the entropy of the adversary's prediction of the sensitive attribute rather than minimizing its accuracy. This approach improves privacy, as it offers the practical benefit of not requiring access to sensitive labels during encoder training. Jaiswal et al. [28] introduced Adversarial Forgetting, a framework that decouples the learning of rich representations from the selective forgetting of sensitive or nuisance information via a dedicated forget-gate mechanism. Building on adversarial fairness frameworks, FR-Train [40] incorporates a mutual information-based formulation and adds a second adversary to enhance robustness to poisoned data.

**Other Approaches**  Recognizing the limitations of adversarial frameworks, particularly their instability and lack of formal guarantees [20, 26, 22], researchers have explored alternative paradigms for learning fair representations. Jiang et al. [29] propose a theoretically grounded approach that enforces demographic parity by minimizing the Wasserstein-1 distance between model outputs across different sensitive groups. Tucker and Shah [47] present Concept Subspace Networks (CSNs), a prototype-based architecture that unifies fair and hierarchical classification within a single model. Ultimately, building on these ideas, Balunović et al. [2] introduce Fair Normalizing Flows (FNF), a framework that provides provable fairness guarantees against any downstream adversary. FNF represents the current state-of-the-art in terms of fair representations. It employs separate normalizing flow encoders for each sensitive group and ensures fairness by minimizing the statistical distance between the resulting latent distributions. This formulation enables exact likelihood computation in the latent space, allowing for theoretical upper bounds on unfairness for any downstream classifier, a property not commonly achieved in existing fair representation learning methods. Furthermore, Balunović et al. [2] show how adversarial learning inherently causes a false sense of fairness [22, 52, 20, 26]. Indeed, they show how multiple methods based on such techniques share a tendency to break once the learned representations are tested on more powerful families of classifiers. Though adversarial learning is the only approach offering specialized representations, such evidence reinforces the need to develop novel directions to achieve them without the instabilities of adversarial learning.

## 3  Background

Let $X \in \mathbb{R}^d$ denote a feature vector, $Y \in \{0, 1\}$ a binary label, and $S \in \{s_1, \ldots, s_n\}$ a sensitive attribute, taken from some joint distribution $\mathbb{P}_{X,Y,S}$. The most common scenario in fairness-critical applications is *binary* sensitive attribute $S \in \{0, 1\}$. Traditional classification algorithms fit a classifier $f_\theta : \mathbb{R}^d \to \{0, 1\}$ to predict the task label $Y$ from $X$. The sensitive attribute $S$ is often statistically correlated with both the feature vector $X$ and the target label $Y$, raising fairness concerns. A prevalent approach in state-of-the-art fairness-aware methods involves constructing fair representations $Z$ from $X$ prior to predicting $Y$.

**Statistical Distance and Adversarial Evaluation**  Since strict fairness criteria often entail accuracy trade-offs due to correlation between the sensitive $S$ and the task label $Y$, fair representations typically result in either performance reductions or leakage of the sensitive information in the classification. Consequently, we adopt the concept of $\epsilon$-*fairness*, where $\epsilon$ represents the statistical distance between the learned representations $\mathbb{P}_{Z|S}$. This enables the application of bounds from Madras et al. [35] relating statistical distances to several fairness metrics (see appendix A.4). Specifically, they consider the Total Variation (TV) distance, defined for distributions $\mathbb{P}_0$ and $\mathbb{P}_1$ as:

$$\Delta(\mathbb{P}_0, \mathbb{P}_1) = \frac{1}{2} \int_{\mathcal{X}} |\mathbb{P}_0(x) - \mathbb{P}_1(x)| \, dx. \tag{1}$$

Balunović et al. [2] estimate the statistical distance directly. However, this can be extremely hard in the general case, and inaccuracy in the estimation might lead to a poor estimation of fairness. Given that the proposed method does not always impose specific distributional assumptions on $Z$, we uniformly adopt *adversarial evaluation* to quantify fairness throughout the whole paper. Indeed, for an optimal adversarial classifier $f$, the statistical distance between conditional distributions can be precisely expressed as:

$$\sup_{\mathcal{G}} \max_{g \in \mathcal{G}} \mathbb{P}(Y = g(X)) = \frac{1}{2} \left(1 + \Delta(\mathbb{P}_{Z|s=0}, \mathbb{P}_{Z|s=1})\right). \tag{2}$$

*Adversarial Evaluation* should not be confused with *Adversarial Learning*. Adversarial Learning learns a latent fair representation $Z$ from $X$ optimizing eq. (2) directly via a min-max training, which can lead to training instability [22, 37, 20]. Adversarial frameworks are typically robust only to the class of functions used during training. Adversarial Evaluation simply assesses the fairness of a fixed representation $Z$ by quantifying how accurately a classifier (often an Multi Layer Perceptron (MLP)) can predict the sensitive attribute $S$ from $Z$.

**Logistic Regression**  Logistic Regression (LR) is a popular classification algorithm used to model the probability of a binary outcome $Y \in \{0, 1\}$ as a function of a set of predictors $X \in \mathbb{R}^d$. The

model is defined by the logistic function $\sigma_\beta : \mathbb{R}^d \to (0, 1)$:

$$\sigma_\beta(x) = \frac{1}{1 + e^{-\beta_{[0]} - \sum_{i=1}^{d} \beta_{[i]} x_{[i]}}}, \tag{3}$$

where the coefficients $\beta \in \mathbb{R}^{d+1}$ are the model parameters, with $\beta_{[0]}$ denoting the bias term. The model output $\sigma_\beta(x)$ is used to learn $\mathbb{P}(Y = 1 \mid X = x)$ via maximum likelihood estimation. To achieve this, the objective function to be minimized is given by the negative log-likelihood:

$$\mathcal{L}^{\text{LR}}(\beta) = -\mathbb{E}_{X,Y}\left[Y \log(\sigma_\beta(X)) + (1 - Y) \log(1 - \sigma_\beta(X))\right] \tag{4}$$

A key advantage of LR over other families of classifiers, such as MLP, is the convexity of its objective function. Since this loss function is convex, it has a global optimum that can be provably reached. This property not only guarantees convergence but also enables the use of second-order optimization techniques such as Newton-Raphson or its variant, Iteratively Reweighted Least Squares. These methods leverage the Hessian of the loss function to achieve quadratic convergence, significantly accelerating the optimization compared to first-order methods like gradient descent.

## 4 Fair Representations matching Characteristic Function

The concept of Characteristic Function has been extensively studied for a long time in the statistical testing literature [33], but interest in its application to ML has only recently grown [1]. As discussed in section 3, to incentivize fair representation, we need to match the different conditional distributions $\mathbb{P}_{Z|S}$, thus minimizing their relative statistical distance. To achieve this, we propose the addition of a differentiable penalty term based on the Characteristic Function Distance. We refer to the overall approach as Fairness matching Characteristic Function (FmCF).

**Characteristic Function Distance** Given a random variable $X \in \mathbb{R}^d$, an alternative way to describe the distribution of $X$ is the Characteristic Function (CF) $\varphi_X : \mathbb{R}^n \to \mathbb{C}$. Denoting by $\mathbb{P}_X$ the probability measure of $X$, the characteristic function $\varphi_X$ is defined as:

---

**Algorithm 1** FmCF loss term

1: **Input:** encoder $h_\theta$, predictor $f_\theta$, batch $\mathcal{B} = \{(x_i, y_i, s_i) \sim \mathbb{P}_{X,Y,S}\}$.
2: $z \leftarrow h_\theta(x)$
3: $\hat{y} \leftarrow f_\theta(z)$
4: **for all** $s \in S$ **do**
5:     **for all** $j \in \{1, \ldots, k\}$ **do**
6:         Sample $t_j \sim \mathbb{P}_T$
7:         $\varphi_\mathcal{N}(t_j) \leftarrow e^{-0.5\|t_j\|^2}$
8:         $\hat{\varphi}_{Z|s}(t_j) = \frac{1}{n} \sum_{i=1}^{n} e^{i\langle t_j, z_i\rangle}$
9:     **end for**
10: **end for**
11: $\mathcal{L}_{\text{CF}} \leftarrow \sum_{s \in \mathcal{S}} \frac{1}{k} \sum_{j=1}^{k} \left|\varphi_\mathcal{N}(t_j) - \hat{\varphi}_{Z|s}(t_j)\right|^2$
12: $\mathcal{L} \leftarrow \mathcal{L}_C(\hat{y}, y) + \alpha \, \mathcal{L}_{\text{CF}}$

---

$$\varphi_X(t) = \mathbb{E}_X\left[e^{i\langle X, t\rangle}\right] = \int_{x \in \mathbb{R}^d} e^{i\langle x, t\rangle} d\,\mathbb{P}_X. \tag{5}$$

The CF is closely related to the notion of the Fourier Transform, and inherits several important properties. It always exists, it's bounded $|\varphi_X(t)| \leq 1$, but most importantly, there is a one-to-one correspondence between probability measures and CFs [33]. That is, given two random variables $X$ and $Y$, with probability measures $\mathbb{P}_X$ and $\mathbb{P}_Y$ then it holds that:

$$\varphi_X = \varphi_Y \iff \mathbb{P}_X = \mathbb{P}_Y. \tag{6}$$

The CF can be used to define a distance metric between probability measures [1]. Given two random variables $X \in \mathbb{R}^d, Y \in \mathbb{R}^d$ with probability measures $\mathbb{P}_X$ and $\mathbb{P}_Y$, the CFD is defined as:

$$\text{CFD}^2_{\mathbb{P}_T}(\mathbb{P}_X, \mathbb{P}_Y) = \mathbb{E}_T\left[|\varphi_X(T) - \varphi_Y(T)|^2\right]. \tag{7}$$

The value of $\text{CFD}_{\mathbb{P}_T}$ depends on the weighting kernel $\mathbb{P}_T$. When $\text{supp}(\mathbb{P}_T) = \mathbb{R}^d$, then the CFD is strict, in the sense that $\text{CFD}_{\mathbb{P}_T}(\mathbb{P}_X, \mathbb{P}_Y) = 0 \iff \mathbb{P}_X = \mathbb{P}_Y$. Common choices for $\mathbb{P}_T$ that ensure this property are the Normal and Laplace distributions.

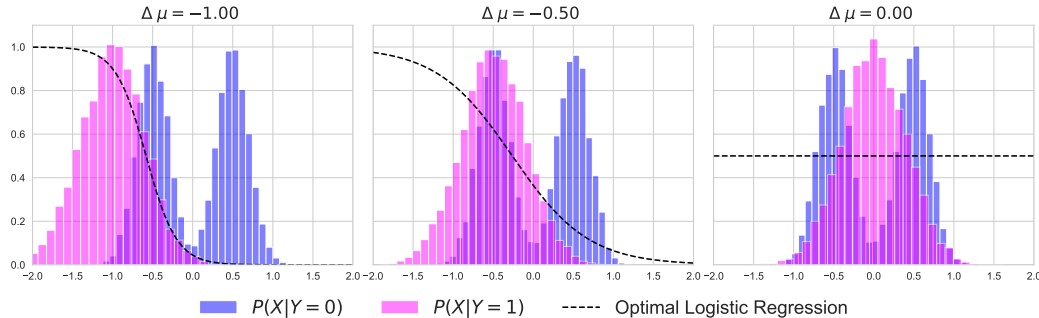

Figure 2: The closer $\mathbb{E}[X|y=0]$ is to $\mathbb{E}[X|y=1]$, the less predictive power the feature has, causing the LR coefficient $\beta$ to approach 0.

**Derivation of Penalty Term**    Instead of matching $\mathbb{P}_{Z|S}$ to each other, each $\mathbb{P}_{Z|S}$ is matched independently to a common target distribution. This simplifies and stabilizes the training procedure.

In this work, we use a standard Normal distribution as the target, although in principle, any distribution could be used. The CF of the standard Normal distribution $\varphi_{\mathcal{N}}$ is given by:

$$\varphi_{\mathcal{N}}(t) = \int_{x \in \mathbb{R}^d} (2\pi)^{\frac{-d}{2}} e^{-\frac{\|x\|^2}{2}} e^{i\langle x, t\rangle} dx = e^{-\frac{\|t\|^2}{2}}. \tag{8}$$

The CFD between $\mathbb{P}_{Z|S}$ and the target distribution can be estimated using Monte Carlo sampling. Given a batch of i.i.d. samples $z_1, \ldots, z_n$ taken from $\mathbb{P}_{Z|S}$ and $t_1, \ldots, t_j$ taken from the weighting kernel $\mathbb{P}_T$ the CFD can be approximated as:

$$\mathrm{CFD}^2_{\mathbb{P}_T}\left(\mathbb{P}_{Z|S}, \mathbb{P}_{\mathcal{N}}\right) \approx \frac{1}{k} \sum_{j=1}^{k} \left|\varphi_{\mathcal{N}}(t_j) - \hat{\varphi}_{Z|S}(t_j)\right|^2. \tag{9}$$

where $\hat{\varphi}_{Z|S}(t) = \frac{1}{n} \sum_{i=1}^{n} e^{i\langle t, z_i\rangle}$ denotes the empirical CF. The total penalty term is obtained by summing the CFD estimate across all different sensitive groups $\mathcal{S}$

$$\mathcal{L}^{\mathrm{CF}} = \sum_{s \in \mathcal{S}} \frac{1}{k} \sum_{j=1}^{k} \left|\varphi_{\mathcal{N}}(t_j) - \hat{\varphi}_{Z|S=s}(t_j)\right|^2. \tag{10}$$

**Comparison to Adversarial Learning and FNF**    Similarly to FmCF, Adversarial Learning offers highly specialized representations; however, they have been shown to provide only an illusory sense of fairness [37, 52, 45], primarily due to the inherent characteristics of the underlying optimization problem. In contrast, the proposed approach seeks to learn representations that are strongly predictive of $Y$ while minimizing information about $S$, without the need for adversarial losses.

Compared to FNF [2], FmCF offers multiple advantages compared to Normalizing Flows (NF). Firstly, NFs require carefully designed architectures. To efficiently compute the Jacobian determinant, only specific layer structures are suitable. Additionally, to ensure invertibility, the dimensionality of the input vector $X$ must be preserved throughout the transformation. This constraint makes NFs computationally expensive for high-dimensional data. In contrast, for FmCF, any function approximator can serve as an encoder. This architectural flexibility enables the learned representation $Z$ to focus solely on the information pertinent to the downstream task (e.g., classification). Moreover, FmCF eliminates the need for sensitive information during deployment. Since the classifier internally debiases $X$, it does not require access to the sensitive attribute $S$ at test time. On the other hand, FNF not only necessitates access to the sensitive attribute but also requires training a separate Normalizing Flow for each sensitive class, significantly increasing training costs. This distinction is particularly important for minimizing potential discrimination and addressing concerns related to the explicit collection of sensitive data.

# 5 Fair Classification matching Sufficient Statistics

The method introduced in section 4 provides a versatile framework for learning specialized representations $Z$ from $X$ independent from $S$ for a general task. However, in practice, most fairness-aware scenarios involve classification [50, 34, 54]. By simplifying the approach, tailoring it to fair classification, we can provide provable post-hoc fairness guarantees. Furthermore, we can also make it computationally cheaper, relaxing the need for a Monte Carlo estimate of the CF.

**Algorithm 2** FmSS Training

---
1: **Input:** encoder $h_\theta$, linear regression $f_\beta$, batch $\mathcal{B} = \{(x_i, y_i, s_i) \sim \mathbb{P}_{X,Y,S}\}$.
2: $z \leftarrow h_\theta(x)$
3: $\hat{y} \leftarrow f_\beta(z)$
4: **for all** $s \in S$ **do**
5: $\quad \sigma_s^2 \leftarrow \text{Var}[Z \mid S = s]$
6: $\quad \mu_s \leftarrow \mathbb{E}[Z \mid S = s]$
7: **end for**
8: $\mathcal{L}_{\text{KL}} \leftarrow \sum_{s \in \mathcal{S}} ||\sigma_s^2 + \mu_s^2 - 1 - \log \sigma_s^2||_1$
9: $\mathcal{L} \leftarrow \mathcal{L}_C(\hat{y}, y) + \alpha \, \mathcal{L}_{\text{KL}}$

---

**Connection to Distribution Moments** The penalty term introduced in section 4 involves sampling multiple points $t \sim \mathbb{P}_T$ from the weighting kernel where the empirical CF is evaluated at. The number of points needed for a reliable estimate of the CFD grows rapidly in the number of dimensions [1].

In this section, we propose an alternative to reduce the number of points needed for the evaluation. Consider the multidimensional Maclaurin expansion of the CF:

$$\varphi_X(t) \approx \sum_{n=0}^{\infty} \sum_{\substack{n_1,\ldots,n_d \\ \text{s.t. } \sum_i n_i = n}}^{\infty} \frac{t_{[1]}^{n_1} \cdots t_{[d]}^{n_d}}{n_1! \cdots n_d!} \left( \frac{\partial^n \varphi_X}{\partial t_{[1]}^{n_1} \cdots \partial t_{[d]}^{n_d}} \right) \Big|_{t=0}. \tag{11}$$

Assuming that $\varphi_X$ is analytic, matching the derivatives evaluated at $t = 0$ would be sufficient to ensure fairness, without the need for sampling from $\mathbb{P}_T$. There exist statistical tools to estimate the full series (appendix A.3); however, they might lead to strong instabilities in training.

A pragmatic approach is to truncate the expansion to some order $N$. This objective does not guarantee that the two distributions are exactly equal; however, we will show that truncating to $N = 2$ is sufficient to ensure fair classification.

In fact, there is a direct connection between the derivatives of the CF and the moments of a distribution.

**Theorem 5.1.** *The $n$-derivative of the CF evaluated at $t = 0$ is related to the $n$-th moment of the distribution:*

$$\frac{\partial^n \varphi_X}{\partial t_{[k_1]}, \ldots, \partial t_{[k_n]}} \Big|_{t=0} = i^n \, \mathbb{E}_X \left[ X_{[k_1]} \ldots X_{[k_n]} \right]. \tag{12}$$

Similarly, the $n$-th empirical moment is equivalent to the $n$-th derivative of the empirical CF.

**Simplified Penalty for Classification** A generic MLP with $M$ layers used for classification can be interpreted as $M - 1$ layers of encoder $z = h_\theta(x)$, and a final layer of a LR $\hat{y} = \sigma_\beta(z)$. Therefore, by applying a fairness penalty to $z$, we can restrict the analysis to the family of LR classifiers.

There exists an optimal condition under which LR is provably fair. In particular, if the first moment $\mathbb{E}[Z \mid S]$ is the same for all $s \in S$, then such a representation $Z$ has no predictive power for LR (i.e. $\beta = 0$) in predicting $S$.

**Theorem 5.2.** *For a representation $Z \in \mathbb{R}^d$ and a binary sensitive attribute $S \in \{0, 1\}$, the optimal LR classifier $\sigma_{\beta^*}$ with $S$ as target is invariant to $Z_{[i]}$ when $\mathbb{E}_{Z|S}\left[Z_{[i]}\right] = \mathbb{E}_Z\left[Z_{[i]}\right]$.*

Theorem 5.2 shows that, building an intermediate representation $Z$ with $\mathbb{E}_{Z|S}[Z] = \mathbb{E}_Z[Z]$ is sufficient to ensure that LR models do not use information about the sensitive attribute $S$.

However, it is possible to derive an even stronger guarantee, analyzing what happens for $\mathbb{E}_Z\left[Z_{[i]} \mid S\right] \approx \mathbb{E}_Z\left[Z_{[i]}\right]$. In particular, theorem 5.3 highlights that the mean and variance of $\mathbb{P}_{Z|S}$ are enough to fully characterize the predictive power of a feature.

**Theorem 5.3.** *The predictive power of a feature $Z_{[i]}$ goes to zero the more its conditional expectation is close to the marginal one. That is, given $\Omega = \dfrac{\left\| \mathbb{E}_{Z|S}\left[Z_{[i]}\right] - \mathbb{E}_Z\left[Z_{[i]}\right] \right\|^2}{\mathbb{E}_Z\left[Z_{[i]}^2\right]}$*

$$\lim_{\Omega \to 0} \beta_{[i]}^* = 0 \tag{13}$$

One empirical example of the consequences of theorem 5.3 is shown in fig. 2 where a Gaussian and a mixture of Gaussians are used to model $\mathbb{P}_{Z|S}$, and $\mathbb{E}_{Z|S}[Z]$ gradually converge to the same value, while the variance is preserved.

Similarly to section 4 we adopt a Gaussian target distribution for $\mathbb{P}_{Z|S}$ This is a particularly compelling choice from the lens of the maximum entropy principle[2].

The fairness penalty term can be obtained from the Kullback–Leibler (KL) divergence between a standard Gaussian $\mathcal{N}(0,1)$ and $\mathcal{N}(\mu, \sigma^2)$, which is given by

$$\mathrm{KL}(\mu, \sigma) = \frac{1}{2}\left(\sigma^2 + \mu^2 - 1 - \log \sigma^2\right).$$

Accordingly, the total penalty term can be obtained as

$$\mathcal{L}^{\mathrm{SS}} = \sum_{s \in \mathcal{S}} \|\mathrm{Var}[Z \mid S = s] + \mathbb{E}\left[Z \mid S = s\right] - 1 - \log\left(\mathrm{Var}[Z \mid S = s]\right)\|. \tag{14}$$

where $\mathrm{Var}[Z \mid S]$ and $\mathbb{E}\left[Z \mid S\right]$ denote the empirical moments.

**Fairness Guarantees**  This approach fundamentally departs from popular methods in the literature, which primarily focus on minimizing $\Delta(\mathbb{P}_{Z|S=0}, \mathbb{P}_{Z|S=1})$, by permitting the sensitive attribute $S$ to be encoded within the learned representation. Crucially, it ensures that $S$ is embedded in a manner that renders it provably inaccessible to the classifier being trained. Leveraging the convexity of LR, one can precisely quantify the $\epsilon$-fairness of a given representation $z$, thus establishing that no logistic classifier $\sigma_\beta(x)$ can extract more than an $\epsilon$ amount of information regarding the sensitive attribute $s$. This constitutes a substantially stronger guarantee than adversarial evaluation, which lacks provable tightness and robustness.

## 6  Experimental Evaluation

FNF [2] offers the strongest theoretical and empirical guarantees, making it the current state of the art in fair representation learning. Therefore, we primarily evaluate the performance of our proposed methods on the same suite of benchmarks, while also considering additional methods and datasets for a more comprehensive analysis. In all tested datasets, both proposed approaches match or surpass the performances of state-of-the-art approaches in terms of accuracy, while providing fairer representations. Surprisingly, even though the approach presented in section 5 gives guarantees against the worst-case linear attacker, it still holds good performances against adversarial evaluations using deep neural networks. Other results on different setups can be found in appendix A.5, where we additionally report more Pareto plots using different hyperparameters for both FmSS (section 5) and FmCF (section 4)

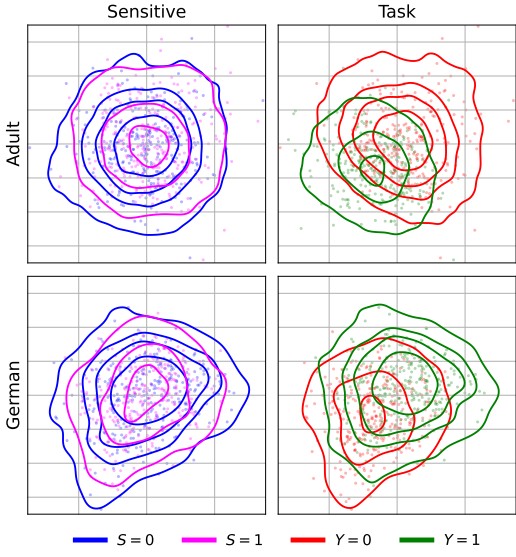

Figure 3: Latent distributions of $\mathbb{P}_{Z|S}$ and $\mathbb{P}_{Z|Y}$ obtained using FmCF.

---

[2]When constrained by fixed first and second moments, the Gaussian distribution uniquely maximizes entropy.

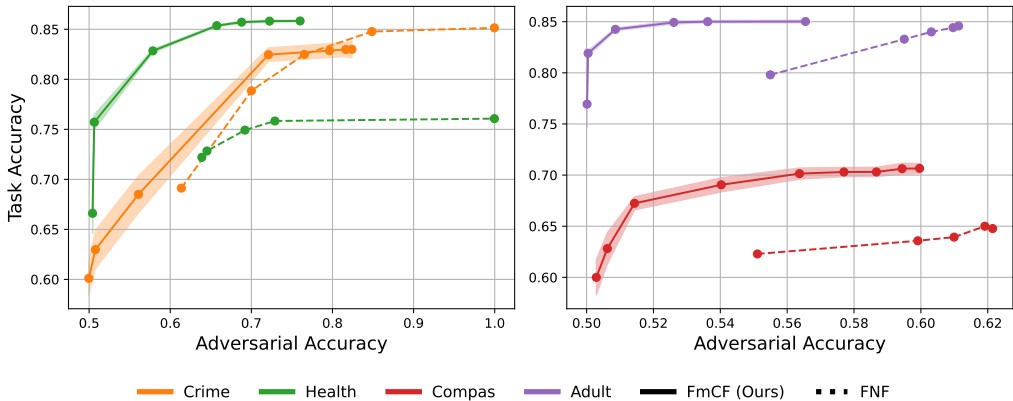

Figure 4: Pareto front between FmCF and FNF comparing task accuracy and fairness (95% confidence intervals from 10 seeds).

**Adult Dataset** This dataset is the most fundamental benchmark in fairness, as almost all methods in literature are evaluated on it [28, 41, 35, 2]. It requires predicting the income of a person without considering their gender [14]. For this dataset, we use the same setup as [2], by evaluating our approach both on a classifier in the same family as the classifier used for the task, and one from a family of much more expressive ones, thus deeper and wider. This allows for a much tighter evaluation of the statistical distance $\Delta(\mathbb{P}_{Z|S=0}, \mathbb{P}_{Z|S=1})$. However, in appendix A.5 we compare to other baselines that only offer guarantees on the trained family

Table 1: Comparison on Adult dataset.

| Model | Acc. | Adv. Accuracy | |
| --- | --- | --- | --- |
| | | $g \in \mathcal{G}$ | $g \notin \mathcal{G}$ |
| Random | 75.20 | 50.00 | 50.00 |
| AdvForgetting [28] | 85.99 | 66.68 | 74.50 |
| MaxEnt-ARL [41] | 84.80 | 69.47 | 85.18 |
| LAFTR [35] | 86.09 | 72.05 | 84.58 |
| FNF [2] | 84.43 | N/A | 59.56 |
| FmCF (ours) | 85.01 | 54.92 | 56.64 |
| FmSS (ours) | 85.02 | 51.70 | 57.47 |

of functions. For the proposed methods in table 1 $g \in \mathcal{G}$ is the class of LR classifiers, while $g \notin \mathcal{G}$ are deep MLPs (see appendix A.7 for details). By construction, FmCF should be evaluated for $g \notin \mathcal{G}$, while FmSS for $g \in \mathcal{G}$.

**German dataset** The German dataset has been explored in the literature under different settings. In particular, [37] considers gender as the sensitive attribute, with the task of predicting income based on a set of personal features. To ensure a fair comparison, we evaluate our approach against several existing methods using the same experimental setup as in [37], as presented in table 2. The proposed method outperforms the other approaches both in terms of accuracy and fairness, showcasing the effectiveness of the novel penalties. Further comparison under sensitive attributes is presented in appendix A.5.

Table 2: Comparison on German dataset.

| Model | Acc. | Adv. Acc. |
| --- | --- | --- |
| Random | 69.00 | 70.00 |
| VFAE [32] | 72.00 | 71.70 |
| CIAFL [32] | 69.50 | 81.10 |
| IRwAL [37] | 71.00 | 69.80 |
| FmCF (ours) | 74.60 | 69.00 |
| FmSS (ours) | 74.20 | 69.00 |

**Comparison with FNF [2]** Since [2] achieves the best balance between fairness and accuracy, in section 6 we compare our proposed method from section 4 against all datasets used in their evaluation. These include four distinct datasets: Crime and Health, which contain continuous features that NFs can handle directly, and Compas and Adult, which include discrete features requiring separate handling by Normalizing Flows.

The plots demonstrate that our proposed method consistently outperforms or matches [2], with the added advantage of not requiring the sensitive attribute during evaluation. Additionally, in appendix A.5, we report the Pareto fronts obtained by varying the latent dimension size, highlighting the impact on performance.

**Visualization of Learned Representations** In order to have a better sense of what the CF approach presented in section 4 is learning, we visualize the learned representation in fig. 3.

Indeed, it can be seen that for both datasets, the distribution $\mathbb{P}_{Z|S}$, which is the distribution of interest to evaluate the fairness of a representation, is very close to being a Gaussian. Thus, the CF is effectively enforcing the similarity to such a distribution. Instead, $\mathbb{P}_{Z|Y}$, which instead is the distribution of interest learn the downstream task, does not resemble a Gaussian distribution, and in particular, while $\mathbb{P}_{Z|S=0} \approx \mathbb{P}_{Z|S=1}$, $\mathbb{P}_{Z|Y=0} \not\approx \mathbb{P}_{Z|Y=1}$.

## 7 Conclusions

In this work, we introduce a novel framework for learning fair specialized representations using CFD, addressing key limitations of existing adversarial and normalizing flow-based approaches. By leveraging CFs, our method ensures stable and efficient fairness by minimizing sensitive information leakage while maintaining high predictive performance. We also present a simplified version of the framework that allows for certifiable bounds on the amount of sensitive information used by downstream tasks. Experimental results show that our approaches consistently outperform or match state-of-the-art methods in terms of the fairness-accuracy trade-off across a range of benchmark datasets. Unlike many existing approaches, our method does not require access to sensitive attributes at inference time, making it more practical for real-world applications. Our approach opens a new avenue for research in fair representation learning by offering a principled, non-adversarial alternative to existing methods. However, despite its simplicity and effectiveness, the proposed framework has certain limitations. Similar to many fairness-oriented algorithms, it encounters challenges in high-dimensional settings, where the CFD tends to lose effectiveness. Moreover, the moment-based approach is inherently tailored to classification tasks, leaving opportunities for extending such approaches to regression contexts. These limitations underscore the need for further research to improve the scalability and robustness of our method in such scenarios. Additionally, current fairness benchmarks largely focus on low-dimensional datasets, limiting the ability to comprehensively evaluate and compare methods in more realistic and complex settings. Developing more sophisticated, high-dimensional fairness datasets would greatly enhance the evaluation of approaches like ours and drive further innovation in the field.

### Broader Impact

This research utilizes the Adult, COMPAS, German, Crime, and Health datasets, each of which is extensively acknowledged as a benchmark in the domain of machine learning fairness. Our study is conducted with stringent adherence to ethical standards and a commitment to transparency. By meticulously employing responsible methodologies, we strive to make substantive contributions to the advancement of AI ethics. This work underscores our dedication to fostering fairness and promoting socially responsible practices within the broader machine learning community.

### Acknowledgements

We thank the anonymous reviewers for their insightful comments and suggestions, which helped improve the clarity and quality of this work. We additionally thank Alessandro Fabris and Davide Dalle Pezze, for the valuable early-stage feedback and discussions on this paper.

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

# A Appendix

This appendix provides additional theoretical derivations, algorithmic details, and extended empirical results to support the main paper. It is organized as follows:

- **Appendix A.1**: Complete derivations for Theorems 5.1, 5.2, and 5.3.
- **Appendix A.2**: Analysis of MMD, its relation to characteristic-function distances, and links to adversarial representation learning.
- **Appendix A.3**: Discussion of full moment-matching, the Russian-Roulette estimator for unbiased series truncation, and practical considerations.
- **Appendix A.4**: Formal definitions and commentary on Demographic Parity, Equalized Odds, and other group-fairness criteria.
- **Appendix A.5**: Extended comparisons against baselines, Pareto-front ablations over latent dimensions, and robustness checks.
- **Appendix A.6**: Detailed descriptions of all benchmark datasets, preprocessing pipelines, and protected attribute encoding.
- **Appendix A.7**: Implementation specifics, optimizer settings, hyperparameters, and hardware configuration.

## A.1 Proofs of Theorems

*Proof.* of theorem 5.1

Taking the derivative inside the expectation

$$\frac{\partial^n \varphi_X}{\partial t_{[k_1]}, \ldots, \partial t_{[k_n]}} = \frac{\partial^n \mathbb{E}_X\left[e^{i\langle X, t\rangle}\right]}{\partial t_{[k_1]}, \ldots, \partial t_{[k_n]}} \tag{15}$$

$$= \mathbb{E}_X\left[\frac{\partial^n e^{i\langle X, t\rangle}}{\partial t_{[k_1]}, \ldots, \partial t_{[k_n]}}\right] \tag{16}$$

$$= \mathbb{E}_X\left[i^n X_{[k_1]} \ldots X_{[k_n]} e^{i\langle X, t\rangle}\right]. \tag{17}$$

Evaluating both sides at $t = 0$ concludes the proof: $\qquad\square$

*Proof.* of theorem 5.2

Defining $x_{[0]} = 1$, the derivative of the logistic function can be expressed as

$$\frac{\partial \sigma_\beta(x)}{\partial \beta_{[i]}} = x_{[i]}\, \sigma_\beta(x)(1 - \sigma_\beta(x)). \tag{18}$$

The derivative of the log-likelihood function in eq. (4) can be expressed as

$$\frac{\partial \mathcal{L}^{\mathrm{LR}}(\beta)}{\partial \beta_{[i]}} = -\frac{\partial}{\partial \beta_{[i]}} \mathbb{E}_{X,Y}\left[Y \log(\sigma_\beta(X)) + (1 - Y)\log(1 - \sigma_\beta(X))\right] \tag{19}$$

$$= -\mathbb{E}_{X,Y}\left[Y \frac{1}{\sigma_\beta(X)} \frac{\partial \sigma_\beta(X)}{\partial \beta_{[i]}} - (1 - Y)\frac{1}{(1 - \sigma_\beta(X))} \frac{\partial \sigma_\beta(X)}{\partial \beta_{[i]}}\right] \tag{20}$$

$$= -\mathbb{E}_{X,Y}\left[Y x_{[i]}\,(1 - \sigma_\beta(x)) - (1 - Y)\,x_{[i]}\,\sigma_\beta(x)\right] \tag{21}$$

$$= -\mathbb{E}_{X,Y}\left[x_{[i]}\,(Y - \sigma_\beta(x))\right]. \tag{22}$$

At the optimum $\beta^*$, the gradient of the log-likelihood is zero. Considering the partial derivative with respect to $\beta_{[0]}$

$$0 = \frac{\partial \mathcal{L}^{\mathrm{LR}}(\beta^*)}{\partial \beta_{[0]}} = -\mathbb{E}_{X,Y}\left[Y - \sigma_{\beta^*}(X)\right] \tag{23}$$

$$= -\mathbb{E}_Y\left[Y\right] + \mathbb{E}_X\left[\sigma_{\beta^*}(X)\right] \tag{24}$$

showing that $\mathbb{E}_Y[Y] = \mathbb{E}_X[\sigma_{\beta^*}(X)]$.

Considering the partial derivative with respect to $\beta_{[i]}$ for $i \neq 0$

$$0 = \frac{\partial \mathcal{L}^{\text{LR}}(\beta^*)}{\partial \beta_{[i]}} = -\mathbb{E}_{X,Y}\left[X_{[i]}\left(Y - \sigma_{\beta^*}(X)\right)\right] \tag{25}$$

$$= \mathbb{E}_X\left[X_{[i]}\,\sigma_{\beta^*}(X)\right] - \mathbb{E}_Y\left[Y\,\mathbb{E}_{X|Y}\left[X_{[i]}\right]\right]. \tag{26}$$

Substituting $\mathbb{E}_X[\sigma_{\beta^*}(X)] = \mathbb{E}_Y[Y]$ and the hypotesis $\mathbb{E}_{X|Y}\left[X_{[i]}\right] = \mathbb{E}_X\left[X_{[i]}\right]$

$$0 = \frac{\partial \mathcal{L}^{\text{LR}}(\beta^*)}{\partial \beta_{[i]}} = \mathbb{E}_X\left[X_{[i]}\,\sigma_{\beta^*}(X)\right] - \mathbb{E}_Y[Y]\,\mathbb{E}_X\left[X_{[i]}\right] \tag{27}$$

$$= \mathbb{E}_X\left[X_{[i]}\,\sigma_{\beta^*}(X)\right] - \mathbb{E}_X[\sigma_{\beta^*}(X)]\,\mathbb{E}_X\left[X_{[i]}\right] \tag{28}$$

$$= \text{Cov}\left(X_{[i]}; \sigma_{\beta^*}(X)\right) \tag{29}$$

Unless $X_{[i]}$ is constant, the condition $\text{Cov}\left(X_{[i]}; \sigma_{\beta^*}(X)\right) = 0$ is obtained for $\beta^*_{[i]} = 0$, making $\sigma_{\beta^*}(X)$ independent of $X_{[i]}$. $\qquad\square$

*Proof.* theorem 5.3

Following the same steps of the proof for theorem 5.2, we can show that $\mathbb{E}_X[\sigma_{\beta^*}(X)] = \mathbb{E}_Y[Y]$ and $\mathbb{E}_X\left[X_{[i]}\,\sigma_{\beta^*}(X)\right] = \mathbb{E}_Y\left[Y\,\mathbb{E}_{X|Y}\left[X_{[i]}\right]\right]$.

Therefore, the correlation $\text{Corr}\left(X_{[i]}; \sigma_{\beta^*}(X)\right)$ can be expressed as:

$$\text{Corr}\left(X_{[i]}; \sigma_{\beta^*}(X)\right) = \frac{\mathbb{E}_X\left[\left(X_{[i]} - \mathbb{E}_X\left[X_{[i]}\right]\right)\left(\sigma_{\beta^*}(X) - \mathbb{E}_X[\sigma_{\beta^*}(X)]\right)\right]}{\sqrt{\mathbb{E}_X\left[X_{[i]}^2\right]\mathbb{E}_X\left[\sigma_{\beta^*}^2(X)\right]}} \tag{30}$$

$$= \frac{\mathbb{E}_X\left[X_{[i]}\sigma_{\beta^*}(X)\right] - \mathbb{E}_X\left[X_{[i]}\right]\mathbb{E}_X[\sigma_{\beta^*}(X)]}{\sqrt{\mathbb{E}_X\left[X_{[i]}^2\right]\mathbb{E}_X\left[\sigma_{\beta^*}^2(X)\right]}}. \tag{31}$$

Using the triangle inequality $\mathbb{E}_X\left[\sigma_\beta(X)^2\right] \geq \mathbb{E}_X\left[\sigma_\beta(X)\right]^2$ and substituting $\sigma_\beta(X) = \mathbb{E}_Y[Y]$ and $\mathbb{E}_X\left[X_{[i]}\,\sigma_{\beta^*}(X)\right] = \mathbb{E}_Y\left[Y\,\mathbb{E}_{X|Y}\left[X_{[i]}\right]\right]$:

$$\text{Corr}\left(X_{[i]}; \sigma_{\beta^*}(X)\right) \leq \frac{\mathbb{E}_X\left[X_{[i]}\sigma_{\beta^*}(X)\right] - \mathbb{E}_X\left[X_{[i]}\right]\mathbb{E}_X[\sigma_{\beta^*}(X)]}{\sqrt{\mathbb{E}_X\left[X_{[i]}^2\right]}\mathbb{E}_X[\sigma_{\beta^*}(X)]} \tag{32}$$

$$= \frac{\mathbb{E}_Y\left[Y\,\mathbb{E}_{X|Y}\left[X_{[i]}\right]\right] - \mathbb{E}_X\left[X_{[i]}\right]\mathbb{E}_Y[Y]}{\sqrt{\mathbb{E}_X\left[X_{[i]}^2\right]}\mathbb{E}_Y[Y]} \tag{33}$$

$$= \frac{\mathbb{E}_Y\left[Y\left(\mathbb{E}_{X|Y}\left[X_{[i]}\right] - \mathbb{E}_X\left[X_{[i]}\right]\right)\right]}{\sqrt{\mathbb{E}_X\left[X_{[i]}^2\right]}\mathbb{E}_Y[Y]}. \tag{34}$$

Assuming $\Omega = \frac{\left\|\mathbb{E}_{X|Y}\left[X_{[i]}\right] - \mathbb{E}_X\left[X_{[i]}\right]\right\|^2}{\mathbb{E}_X\left[X_{[i]}^2\right]} < \epsilon$ we can bound the correlation with

$$\text{Corr}\left(X_{[i]}; \sigma_{\beta^*}(X)\right) \leq \sqrt{\epsilon}. \tag{35}$$

Showing that $\lim_{\Omega \to 0} \text{Corr}\left(X_{[i]}; \sigma_{\beta^*}(X)\right) = 0$. When $X_{[i]}$ is not constant, this also implies $\lim_{\Omega \to 0} \beta^*_{[i]} = 0$ $\qquad\square$

## A.2 Connections to Maximum Mean Discrepancy

The Maximum Mean Discrepancy (MMD) is a popular distance measure between distributions. Given a Reproducing Kernel Hilber Space (RKHS) $\mathcal{H}$, it is defined as follows

$$\text{MMD}_{\mathcal{H}}\left(\mathbb{P}_X, \mathbb{P}_Y\right) = \sup_{\|f\|_{\mathcal{H}} \leq 1} \mathbb{E}\left[f(X)\right] - \mathbb{E}\left[f(Y)\right]. \tag{36}$$

**Connections from Characteristic Function** For random variables $X \in \mathbb{R}^d$, $Y \in \mathbb{R}^d$, the MMD can be expressed in terms of the CFs [8].

$$\text{MMD}_{\mathcal{H}}^2\left(\mathbb{P}_X, \mathbb{P}_Y\right) = \int_{\mathbb{R}^d} \left|\varphi_X(t) - \varphi_Y(t)\right|^2 w(t)dt. \tag{37}$$

where the weighting function $w(t)$ is the inverse Fourier Transform of the kernel of $\mathcal{H}$. When $w$ is a probability density function, then the integral can be interpreted as an expectation, and

$$\text{MMD}_{\mathcal{H}}\left(\mathbb{P}_X, \mathbb{P}_Y\right) = \text{CFD}_{\mathbb{P}_T}\left(\mathbb{P}_X, \mathbb{P}_Y\right) \tag{38}$$

**Connections from Sufficient Statistics** The formulation presented in section 5 shares some similarities with Maximum Mean Discrepancy (MMD) used by VFAE. Indeed, MMD states that given two distributions, they are equal as long as the maximum distance between their expected value under a transformation $f$ is zero.

Thus, if $\text{MMD}_{\mathcal{H}}\left(\mathbb{P}_X, \mathbb{P}_Y\right) = 0$ then $\mathbb{P}_X = \mathbb{P}_Y$. This is indeed exploited by Adversarial Learning, which tries to find $f$ by means of a min-max optimization. However, as shown by [2], Adversarial Learning fails as finding such $f$ is extremely hard, due to the fact that the two distributions are constantly changing based on the current $f$. Therefore, if $\text{MMD}_{\mathcal{H}}\left(\mathbb{P}_X, \mathbb{P}_Y\right) \neq 0$, no provable guarantee on the fairness can be given. Instead, if $f$ is in the family of LR, theorem 5.3 shows that the second moment bounds the predictive power of such a representation.

This is a weaker condition than the one that [2], Adversarial Learning, CVAE, uses to learn, by matching the distribution. Indeed, $\mathbb{P}_{Z|S=0} = \mathbb{P}_{Z|S=1} \Rightarrow \mathbb{E}[Z \mid S = 1] = \mathbb{E}[Z \mid S = 0]$ but not the other way around.

Yet, given LR has a convex loss function, the optimal adversarial classifier can be provably trained to convergence with a second-order optimizer, thus giving guarantees on the amount of sensitive $S$ that can at most be used in the classification.

## A.3 Matching All Moments

The approaches presented in sections 4 and 5 can be reconducted to matching the moment generating function, or directly the moments of a distribution. Indeed, matching all moments of two distributions with bounded support ensures that they are equal almost everywhere. Even though it is trivial to force the support of $Z$ to be bounded, directly applying moment matching to the distributions $\mathbb{P}_{Z|S}$ might be impractical, as all moments are needed to characterize a distribution, which is computationally unfeasible. Furthermore, truncating to the $k$-th moment leads to a biased estimation, and, in the context of this work, potential leakage of the sensitive attribute $S$.

One powerful tool for unbiased estimation of the complete infinite series of moments is the *Russian Roulette estimator*. Consider an infinite series of the form $Y = \sum_{i=1}^{\infty} Y_i$. The Russian Roulette estimator randomly chooses a truncation point $N$ and calculates the partial sum up to $N$, while adjusting the weight of the computed sum to ensure that the estimator remains unbiased:

$$\hat{Y} = \sum_{i=1}^{N} \frac{Y_i}{P(N \geq i)} \tag{39}$$

While the Russian Roulette estimator is unbiased, it can result in high variance if the truncation probabilities are not chosen carefully. Therefore, balancing computational efficiency and variance is essential in practical applications.

Instead, the proposed approaches presented in sections 4 and 5 use a more principled and computationally stable way of matching two distributions.

## A.4 Additional Fairness Metrics

Although various fairness criteria have been proposed, balancing the fairness-accuracy trade-off remains challenging.

Demographic Parity (DP) is a widely used group fairness criterion that requires the outcome of a decision-making process to be independent of a sensitive attribute such as race, gender, or age [18, 13]. Let $h_\theta$ denote a representation function that maps inputs $X$ to a latent space $Z = h_\theta(X)$, and let $f_\theta$ denote a classifier applied to this representation, producing predictions $\hat{Y} = f_\theta(h_\theta(X))$. Then, DP requires the probability of receiving a favorable outcome (e.g., $\hat{Y} = 1$) to be the same across all demographic groups $s \in \mathcal{S}$:

$$\mathbb{P}(\hat{Y} = 1 \mid S = s) = \mathbb{P}(\hat{Y} = 1) \quad \forall s \in \mathcal{S}. \tag{40}$$

This states that the positive outcome rate for any group $s$ should be equal to the overall positive outcome rate in the population. Since demographic parity is independent of the ground truth labels, it is especially salient in contexts where reliable ground truth information is hard to obtain and a positive outcome is desirable, including employment, credit, and criminal justice [16, 23].

To quantify deviations from this ideal, we measure the Demographic Parity Difference (DPD). In binary group settings (e.g., involving an advantaged group $a$ and a disadvantaged group $d$), it is defined as:

$$\text{DPD} = \mathbb{P}(\hat{Y} = 1 \mid S = a) - \mathbb{P}(\hat{Y} = 1 \mid S = d). \tag{41}$$

In real-world scenarios, sensitive attributes are often multi-class (e.g., race with more than two categories). In these cases, generalizations for measuring DPD violation include:

- The *Maximum Pairwise Difference*, capturing the largest disparity in positive outcome rates between any two groups:

$$\text{DPD} = \max_{s_i, s_j \in \mathcal{S}} \left| \mathbb{P}(\hat{Y} = 1 \mid S = s_i) - \mathbb{P}(\hat{Y} = 1 \mid S = s_j) \right|. \tag{42}$$

- The *Average Absolute Pairwise Difference*, computing the average disparity across all pairs:

$$\text{DPD} = \frac{1}{|\mathcal{S}|^2} \sum_{s_i \in \mathcal{S}} \sum_{s_j \in \mathcal{S}} \left| \mathbb{P}(\hat{Y} = 1 \mid S = s_i) - \mathbb{P}(\hat{Y} = 1 \mid S = s_j) \right|. \tag{43}$$

Since achieving perfect demographic parity (where all the above differences are zero) can be impractical, the objective is often relaxed to minimizing the Demographic Parity Distance:

$$\Delta^{\text{DP}} = \sum_{s \in \mathcal{S}} \left| \mathbb{P}(\hat{Y} = 1 \mid S = s) - \mathbb{P}(\hat{Y} = 1) \right|. \tag{44}$$

While demographic parity is often defined in terms of these outcome rate differences, it can also be understood more generally in terms of statistical distance between group-conditional distributions of model predictions or representations [35]. Let $Z_0$ and $Z_1$ denote the distributions of these representations conditioned on different sensitive attribute values $S = 0$ and $S = 1$, respectively. Then, for any measurable test function $f_\theta$ applied to the model outputs, one can define the test discrepancy as:

$$\Delta^{\text{DP}}(f_\theta \circ h_\theta) = \left\| \mathbb{E}_{Z_0}[f_\theta(Z)] - \mathbb{E}_{Z_1}[f_\theta(Z)] \right\|. \tag{45}$$

The statistical distance (or total variation distance) between $Z_0$ and $Z_1$ is the supremum of this discrepancy over all measurable test functions $\mu$:

$$\Delta^*(Z_0, Z_1) = \sup_{\mu} \left\| \mathbb{E}_{Z_0}[\mu(Z)] - \mathbb{E}_{Z_1}[\mu(Z)] \right\|. \tag{46}$$

It follows that for any $f_\theta$,

$$\Delta^{\text{DP}}(f_\theta \circ h_\theta) \leq \Delta^*(Z_0, Z_1), \tag{47}$$

and $\Delta^{\text{DP}}(f_\theta \circ h_\theta) = 0$ if and only if $f_\theta(h_\theta(X)) \perp S$, i.e., demographic parity holds.

In contrast to Demographic Parity, the Equal Opportunity (EO) criterion incorporates information about the target variable $Y$ [27]. Specifically, EO requires that the true positive rate (TPR) be equal across all demographic groups. Formally:

$$\mathbb{P}(\hat{Y} = 1 \mid S = s, Y = 1) = \mathbb{P}(\hat{Y} = 1 \mid Y = 1) \quad \forall s \in \mathcal{S}. \tag{48}$$

This fairness criterion is especially salient in domains where false negatives are particularly harmful and reasonably accurate ground truth labels are available, such as in healthcare, criminal justice, and risk assessment [7, 43, 38]. The corresponding EO distance is defined as:

$$\Delta^{\text{EO}}(f_\theta \circ h_\theta) = \|\mathbb{E}_Z[f_\theta(Z) \mid S = 0, Y = 1] - \mathbb{E}_Z[f_\theta(Z) \mid S = 1, Y = 1]\| \tag{49}$$

which can be upper bounded by the objective value of an optimal adversary $g^*$ trained to distinguish between $Z \mid Y = 1$ for different groups [35].

An alternative fairness criterion is Equalized Odds (EOd) [27], which extends Equal Opportunity by requiring that prediction outcomes be conditionally independent of the sensitive attribute given the true label. Formally, EOd demands that for all $s \in \mathcal{S}$:

$$\mathbb{P}(\hat{Y} = 1 \mid Y = 1, S = s) = \mathbb{P}(\hat{Y} = 1 \mid Y = 1) \quad \forall s \in S \tag{50}$$

and

$$\mathbb{P}(\hat{Y} = 1 \mid Y = 0, S = s) = \mathbb{P}(\hat{Y} = 1 \mid Y = 0) \quad \forall s \in S \tag{51}$$

These constraints ensure that both the true positive rate (TPR) and false positive rate (FPR) are equal across all groups [56, 25]. This criterion is particularly applicable in decision-making contexts where reliable ground truth labels are available and both false negatives and false positives incur significant societal or personal costs [12, 53]. The EOd distance is defined as:

$$\Delta^{\text{EOd}}(f_\theta \circ h_\theta) = \|\mathbb{E}_Z[f_\theta(Z) \mid S = 0, Y = 1] - \mathbb{E}_Z[f_\theta(Z) \mid S = 1, Y = 1]\|$$
$$+ \|\mathbb{E}_Z[f_\theta(Z) \mid S = 0, Y = 0] - \mathbb{E}_Z[f_\theta(Z) \mid S = 1, Y = 0]\|. \tag{52}$$

### A.5 Additional empirical evaluations and ablations

In table 3a we compare our proposed approaches to other baselines on the Adult dataset, with the same setup as in [37]. Given that no distinction is made on the family of functions used for the evaluation, we report the Adv. Accuracy of FmCF and FmSS approaches using a deep and wide MLP. The same holds true for the results reported in table 3b, where we compare out methods to the same setup from [47]

(a) Comparison on Adult dataset using the setup from [37].

| Model | Acc. | Adv. Acc. |
|---|---|---|
| Random | 75.2 | 67.5 |
| VFAE [32] | 84.2 | 88.2 |
| CIAFL [51] | 83.1 | 88.8 |
| IRwAL [37] | 84.2 | 77.6 |
| FmCF (ours) | 85.0 | 67.8 |
| FmSS (ours) | 85.0 | 67.9 |

(b) Comparison on Adult dataset using the setup from [47].

| Model | Acc. | Adv. Acc. |
|---|---|---|
| Random | 70.0 | 81.0 |
| CSN [47] | 73.1 | 81.3 |
| CIAFL [51] | 73.6 | 81.1 |
| VFAE [32] | 72.8 | 81.2 |
| FRTrain [40] | 72.7 | 80.9 |
| WassDB [29] | 72.8 | 81.1 |
| FmCF (ours) | 74.1 | 81.1 |
| FmSS (ours) | 74.2 | 81.1 |

In figs. 5 to 8, we report the tradeoffs between adversarial balanced accuracy and task accuracy. We evaluate FmCF section 4 against an adversarial MLP as reported in appendix A.7. Instead, we evaluate FmSS section 5 both against an MLP and against a LR. In particular, for all images, in black is reported the performance of Fair Normalizing Flows [2], while in blue is reported the performance using 1 dimension, in orange 2 dimensions, and in green 3 dimensions. We observe that the lower the dimension, the easier it is for the penalties to be effective, giving surprising performances for extremely low-dimensional latent spaces. Notably, such a characteristic is not limiting for the

downstream task, as even in the lowest-dimensional setting, both approaches outperform all the baselines.

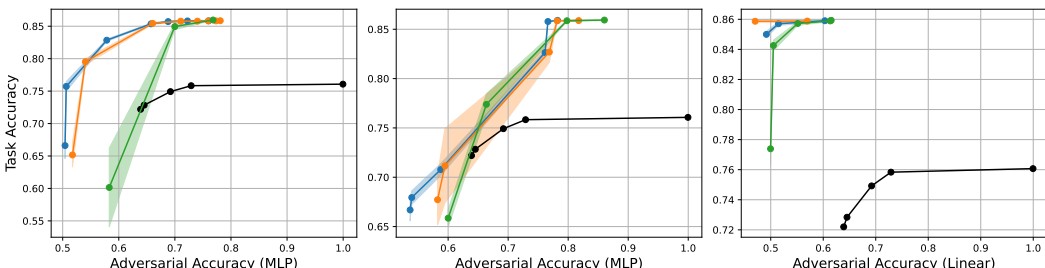

Figure 5: Pareto varying latent dimension on Health dataset.

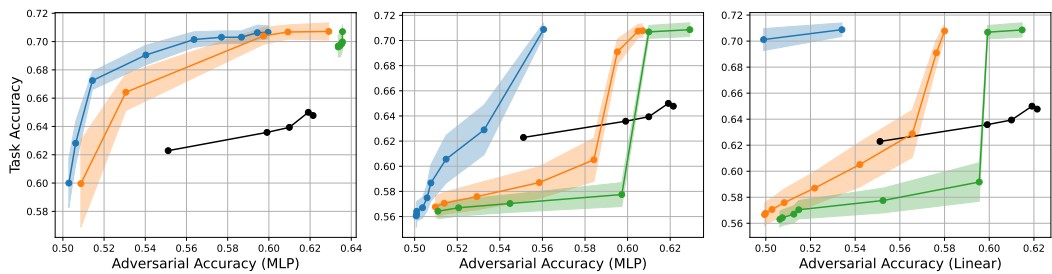

Figure 6: Pareto varying latent dimension on Compas dataset.

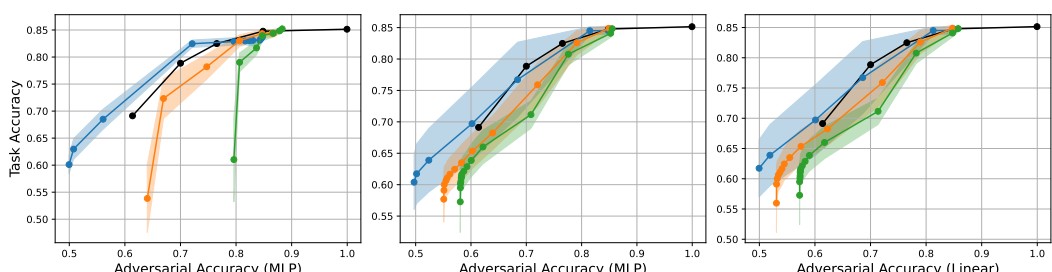

Figure 7: Pareto varying latent dimension on Crime dataset.

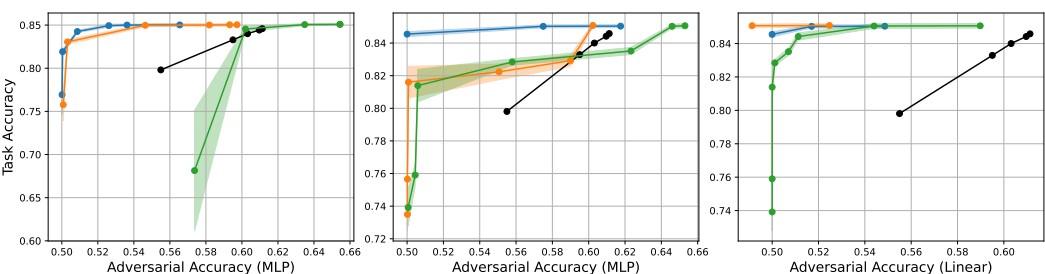

Figure 8: Pareto varying latent dimension on Adult dataset.

Furthermore, we evaluated the effects of batch size on the reliability of the CFD estimation. As confirmed by appendix A.5, larger batches yield more stable and reliable CFD estimates. The variance of the estimates decreases consistently with increasing batch size, aligning with the expected theoretical scaling of $O(\frac{1}{n})$.

Table 4: Standard deviation of FmCF Penalty estimator.

| Batch Size | Adult | German | Compas | Health |
|---|---|---|---|---|
| 8 | 0.1150 | 0.1252 | 0.0924 | 0.1004 |
| 16 | 0.0596 | 0.0854 | 0.0361 | 0.0422 |
| 32 | 0.0164 | 0.0172 | 0.0129 | 0.0121 |
| 64 | 0.0063 | 0.0056 | 0.0052 | 0.0054 |
| 128 | 0.0026 | 0.0024 | 0.0022 | 0.0027 |
| 256 | 0.0016 | 0.0004 | 0.0010 | 0.0015 |
| 512 | 0.0007 | 0.0000 | 0.0007 | 0.0009 |

Given that all benchmarks considered thus far pertain to classification tasks, we now present a simple example demonstrating the applicability of FmCF to alternative problem settings. Specifically, in fig. 9, a conditional convolutional autoencoder is employed to learn a latent representation of the MNIST handwritten digits dataset. Subsequently, the loss function described in algorithm 1 is utilized to penalize the divergence between $\mathbb{P}_{Z|S}$ and a Gaussian distribution. When this condition is satisfied, it ensures that $Z \perp\!\!\!\perp S$, indicating that the latent representation $Z$ encodes information about $X$ that is independent of $S$. Consequently, when the model is prompted to reconstruct an input with a different label, it retains the essential characteristics of the original input while generating a digit consistent with the newly specified label.

Indeed, the decoder is able to generate images preserving characteristics such as rotation, while only relying on the fed label for the kind of digit to show.

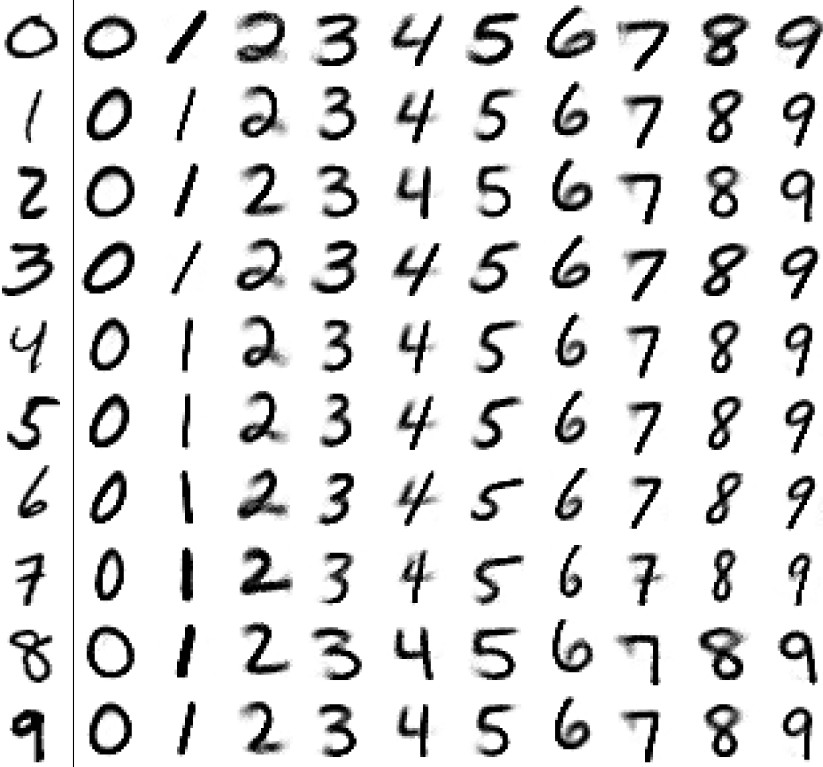

Figure 9: Latent representations $Z$ are extracted from the left-most column and combined with arbitrary target labels to generate stylistically consistent images of different classes, shown on the right.

For more details on the datasets, please refer to appendix A.6.

## A.6 Datasets

We utilize six well-known datasets in our study, sourced from both the UCI Machine Learning Repository and other publicly available resources. These datasets include Adult, Crime, Compas, Law School, Health, and the Statlog (German Credit Data) dataset. Each of these datasets presents distinct challenges and characteristics relevant to fairness and predictive modeling. Below, we briefly introduce each dataset and outline the preprocessing steps employed:

- Adult[3]: The Adult dataset, also known as the Census Income dataset, originates from the 1994 Census database and is available through the UCI repository [17]. It contains 14 attributes, including age, workclass, education, marital status, occupation, race, and sex. The prediction task is to determine whether an individual's income exceeds $50,000 per year. To facilitate modeling, we discretize continuous features and retain categorical variables related to demographics and employment. Sex is treated as the protected attribute in fairness analyses.

- Crime[4]: The Communities and Crime dataset combines socio-economic data from the 1990 US Census, law enforcement data from the 1990 LEMAS survey, and crime data from the 1995 FBI UCR. The goal is to predict whether the violent crime rate of a community is above or below the median. We utilize attributes such as race percentages, income levels, and family structure indicators. Race is designated as the protected attribute, derived from the proportions of racial groups within each community.

- Compas[5]: The Compas dataset contains data related to criminal history, jail and prison time, demographics, and COMPAS risk scores from Broward County (2012–2013) [5]. The prediction task involves forecasting recidivism within two years. Key attributes include age, prior count, and charge degree, with race being the protected attribute. We preprocess the dataset by discretizing continuous variables and retaining categorical ones, critical for risk prediction.

- Health[6]: The Health dataset was part of the Heritage Health Prize competition on Kaggle and contains medical records of over 55,000 patients. We focus on the merged claims, drug count, and lab count attributes while removing personal identifiers to ensure privacy. Age is treated as a protected attribute, divided into binary groups: above and below 60 years. The primary prediction task is to assess the maximum Charlson Comorbidity Index, reflecting the long-term survival prospects of patients with multiple conditions.

- Law School[7]: The Law School dataset consists of data from admissions cycles between 2005 and 2007, covering over 100,000 individual applications. Attributes include LSAT scores, undergraduate GPA, race, gender, and residency status. To enhance privacy, data has been aggregated where necessary. We consider race as the protected attribute, binarizing it into white and non-white categories. The main task is to predict law school admission outcomes.

- German[8]: The German Credit Data dataset, sourced from the UCI Machine Learning Repository [17], consists of 1,000 records of credit applicants. Each instance is labeled as either "good" or "bad" credit risk. The dataset contains 20 features, including age, credit amount, employment status, and personal status, among others. Both categorical and numerical data are present, requiring careful preprocessing. We encode categorical variables using one-hot encoding and normalize numerical features. The protected attribute for fairness evaluation in this dataset is age, segmented into groups representing different age ranges.

However, in [2], the Lawschool dataset is also used to show the effectiveness of FNF. Through the extensive research, we did not find any publicly available version of such a dataset, thus our approaches are not evaluated on it.

## A.7 Training details

All experiments reported in this paper were implemented using PyTorch. The models were trained on a server equipped with an AMD Ryzen Threadripper PRO 5995WX CPU (64 cores, 128 threads),

---

[3]https://archive.ics.uci.edu/dataset/2/adult
[4]https://archive.ics.uci.edu/dataset/183/communities+and+crime
[5]https://github.com/propublica/compas-analysis
[6]https://paperswithcode.com/dataset/heritage-health-prize
[7]https://eric.ed.gov/?id=ED469370
[8]https://archive.ics.uci.edu/dataset/522/south+german+credit

512 GiB of RAM, and three NVIDIA RTX A6000 GPUs. Despite this powerful setup, none of the models utilized more than 4 GiB of VRAM during training.

All reported results are averages over 10 different seeds of the proposed approaches. No extensive hyperparameter tuning was performed for any of the reported results, underscoring the effectiveness of the proposed methods. All MLPs used for adversarial evaluations, encoding, and classification consist of four layers with 64 neurons each. Additionally, larger MLPs for fairness adversarial evaluations, configured with 128 and 256 neurons, were tested but resulted in poorer performance.

The Adam optimizer [30] was employed for all training sessions, with a learning rate of 0.0003. Training was conducted for 100 epochs, incorporating L2 regularization with a weight penalty of 0.0001 to mitigate overfitting. Since accuracy and fairness often involve a trade-off, early stopping was not applied. Instead, the model from the final epoch was used for evaluation.

