# OpenReview forum: "Simple and Effective Specialized Representations for Fair Classifiers"
_NeurIPS.cc/2025/Conference — NeurIPS 2025 poster_

### Official Review · Reviewer_HZGi · 2025-06-19

**Clarity:** 2
**Significance:** 2
**Originality:** 2
**Rating:** 5
**Confidence:** 3

**Summary:**

This paper proposes a representation learning method that overcomes the unstable issues of adversarial learning and computationally intensive issue of distribution matching. The method first uses CFD to align group-wise representations to a common distribution, and then proposes a more efficient moment-matching approximation that matches low-order moments (N = 2) to guarantee that downstream classifiers cannot access sensitive information.

**Questions:**

- How sensitive is the method to the choice of the shared target distribution?
- Have the authors considered whether aligning group-wise representations to a learned barycenter or a task-specific distribution might be more effective than enforcing a shared target distribution, such as the Gaussian used in this paper?
- Line 230: “…however, we show that truncating to N=2 is sufficient to ensure fair classification,” but I cannot find the evidence to support this claim. Can the authors specify and justify it?

**Ethical Concerns:**

["NO or VERY MINOR ethics concerns only"]

**Final Justification:**

The authors have addressed all of my questions and concerns, and I was already inclined to accept the paper from the beginning. In particular, I appreciate the authors’ detailed response regarding the major limitation I raised about the scope of the experimental evaluation. They clarified that although the main focus was on the Adult and German datasets, they aimed to ensure fairness in comparison by restricting evaluations to datasets where shared training configurations were accessible. Additionally, they mentioned a generative task to demonstrate the broader applicability of their CF approach to latent representations, although they deliberately left it out of the main paper due to its divergence from the main classification focus. I find this reasoning acceptable and appreciate the authors’ efforts to maintain fairness and clarity in the empirical study. Taking all of this into account, I have raised my score accordingly.

**Limitations:**

Yes, the authors acknowledge the limitation regarding high-dimensional settings in the conclusion section. However, I respectfully disagree with the claim that "current fairness benchmarks largely focus on low-dimensional datasets, limiting the ability to comprehensively evaluate and compare methods in more realistic and complex settings." In the domain of computer vision, for instance, there exist several well-established image datasets for fairness evaluation, such as CelebA, FairFace, and LFW.

**Paper Formatting Concerns:**

- Equation (2), the left parenthesis in $p(y = f(x))$ seems to have a formatting issue.

- The positions of Algorithm 1 and Algorithm 2 can be adjusted appropriately, especially to ensure that the top and bottom margins are aligned with the surrounding text.

- Lines 242–267: The formatting appears somewhat chaotic, maybe consider reordering or restructuring them in a cleaner and more organized way.

**Quality:**

2

**Strengths And Weaknesses:**

Strengths:
- The proposed method replaces the need to match full distributions (via monte carlo CFD estimation) with a tractable alternative by truncating the characteristic function expansion to a low order (e.g., N = 2), which leads to more stable and efficient training compared to monte carlo based methods.
- The method uses CFD to align group-wise representations at the distributional level, offering a clearer and more principled formulation of fairness compared to adversarial learning or MMD-based approaches.
- The visualization of learned representations in Figure 4 is very intuitive to capture the effectiveness of the proposed method.

Weakness:
- The proposed method aligns representations of each group to a shared target distribution, and in this paper, uses the standard normal distribution (likely for computational simplicity and methodological clarity). However, consider a situation where the target distribution is not well-aligned with any group (e.g., when the data resemble a mixture of Gaussians) and enforcing all groups to align with the shared target may actually distort them unnecessarily.
- One major limitation of the paper lies in its limited experimental scope. The method is only evaluated on two small tabular datasets, which are low-dimensional and structurally simple. In such cases, aligning low-order moments (N=2) may already suffice for fair representation learning. However, it remains unclear whether the proposed approach can generalize to more complex scenarios, such as high-dimensional, nonlinear, or multi-modal tasks, where group-wise distributional differences are more intricate. This limits the method’s relevance to current practice and raises concerns about its applicability in more realistic, large-scale, or multi-modal learning settings.

---

> ### Author Rebuttal · Authors · 2025-07-29
>
> We thank the reviewer for the detailed and constructive feedback. Below, we address each point raised, clarifying theoretical aspects and experimental choices, and acknowledging areas that can be improved in the final version.
>
> Please note that, due to the updated NeurIPS guidelines, we are no longer permitted to upload supplementary materials or a revised version of the paper at this stage. We hope that the clarifications provided here sufficiently address your concerns.
>
> ----
> ### Weaknesses
>
> _"The proposed method aligns representations of each group to a shared target distribution, and in this paper, uses the standard normal distribution (likely for computational simplicity and methodological clarity). However, consider a situation where the target distribution is not well-aligned with any group (e.g., when the data resemble a mixture of Gaussians) and enforcing all groups to align with the shared target may actually distort them unnecessarily."_
>
> The proposed method based on CF aims to make two changes of variable. First $X$, the original sample, is mapped to $Z$, a fair representation, and then $Z$ is mapped to $Y$, the classification. An MLP can provably perform any arbitrary change of variable (thus is able to map $X$ to $Z$, a Gaussian distribution, with no problem) thanks to the universal approximation theorem.
> Indeed, this is the core idea behind every fair representation learning algorithm (VAE maps to a Gaussian, FNF to an estimated distribution, Adversarial learning to two distributions that should be indistinguishable).
> Nonetheless, in order to have guarantees about the expressivity, we just need to ensure that $H(Z) \ge H(Y)$, thus that it has enough capacity to “be able to carry the relevant information for the classification”.
> Since $Z$ is continuous (and no noise is injected) and $Y$ is discrete, we know this is possible. To conclude, though mapping to a distribution might sound tedious for certain cases, an MLP is more than capable of doing so.
> However, if you feel that this discussion (ie, that mapping to a Gaussian is not “theoretically limiting”) needs more attention, let us know and we will make sure to add more details to the camera-ready version of the paper if it gets accepted.
>
> _"One major limitation of the paper lies in its limited experimental scope. The method is only evaluated on two small tabular datasets, which are low-dimensional and structurally simple. In such cases, aligning low-order moments (N=2) may already suffice for fair representation learning. However, it remains unclear whether the proposed approach can generalize to more complex scenarios, such as high-dimensional, nonlinear, or multi-modal tasks, where group-wise distributional differences are more intricate. This limits the method’s relevance to current practice and raises concerns about its applicability in more realistic, large-scale, or multi-modal learning settings."_
>
> Indeed, a thorough evaluation has been carried out on the Adult and German datasets, as they are the benchmarks most commonly used in the literature.
> However, the empirical evaluation is not limited to this two datasets. For instance, Figure 3 also shows results for CRIME, COMPAS, and HEALTH datasets in comparison to FNF [2].
> The reason why other methods were not evaluated on such datasets was to be as fair as possible to the compared methods. Indeed, all the other baselines considered for this study performed their evaluation only on German and Adult, and shared the code and training configuration (network, optimizer, and so on) only for such datasets.
> Instead, FNF also tested their beyond GERMAN and ADULT. Thus, we were able to do a fair comparison with such work.
> Overall, we tried to make our experimental evaluation as broad and comprehensive as any of the baselines we evaluated against.
> Furthermore, while we acknowledge that it might be a simple toy scenario, we felt that showing that the proposed approach is valid on different modalities was important too. Thus, we included a simple generative task in the appendix.
> This showcases the effectiveness of the CF approach for modelling the distribution of latent representations learned by autoencoders.
> Yet, given that the main focus of the paper was to deal with fair classification, we thought that it would have been a bit off-topic to put it in the main text.
>
> ### Question
>
> _"How sensitive is the method to the choice of the shared target distribution?"_
>
> Given that we wanted an effective and simple approach to learn fair representations, we tried to do as minimal hyperparameter tuning as possible. Therefore, we did not test other reference distributions beyond the Gaussian one.
> Ideally, if we consider that an MLP can perform an arbitrary change of variable, we can easily show that we should be invariant to the reference distribution.
> In fact, the network can learn the map $X \rightarrow Z \rightarrow N(0,1) \rightarrow Y$, where $Z$ is modelled with a different distribution.
>
> _"Have the authors considered whether aligning group-wise representations to a learned barycenter or a task-specific distribution might be more effective than enforcing a shared target distribution, such as the Gaussian used in this paper?"_
>
> We did not explore this direction, but the idea of learning the reference distribution might be an interesting line of work, as it would allow the MLP to choose the distribution that is “the simplest”.
> We will make sure to include this option in the future works in the camera-ready version if the paper gets accepted.
>
> _"Line 230: “…however, we show that truncating to N=2 is sufficient to ensure fair classification,” but I cannot find the evidence to support this claim. Can the authors specify and justify it?"_
>
> The result showing that we only need 2 moments is the conjunction of Theorems 5.2 and 5.3. Theorem 5.2 shows that as long as the first moment is exact, such a feature has no predictive power for any LogRegression. Theorem 5.3 shows that if the first moment is an epsilon away between the two sensitive groups, the predictive power of such a feature is bound by its variance. To conclude, if we have bounded variance and we push the means towards each other, we can ensure fairness.
> Line 230 is meant to anticipate these results, but we realize that this could be expressed much more clearly. Thus, we will make sure to revise it in the camera-ready version, if the paper gets accepted.
>
> ### Limitations
>
> _"Yes, the authors acknowledge the limitation regarding high-dimensional settings in the conclusion section. However, I respectfully disagree with the claim that "current fairness benchmarks largely focus on low-dimensional datasets, limiting the ability to comprehensively evaluate and compare methods in more realistic and complex settings." In the domain of computer vision, for instance, there exist several well-established image datasets for fairness evaluation, such as CelebA, FairFace, and LFW."_
>
> We thank the reviewer for the honest comment and for bringing such benchmarks to our attention. Given the presence of such benchmarks, we agree that the claim might be a bit of a stretch. What we meant is that none of the other considered baselines evaluate their approach on such datasets.
> We can include in the appendix an evaluation of the proposed approaches (FmCF, FmSS) on such datasets, in order to highlight that the input modality is not a point of friction.
> In any case, we will make sure to properly address this aspect in the revised paper for the camera-ready version if it gets accepted.
>
>
>
> ### Formatting Concerns
>
> _"Equation (2), the left parenthesis in $p(y=f(x))$ seems to have a formatting issue._
> Thanks for spotting the typo
>
> _"The positions of Algorithm 1 and Algorithm 2 can be adjusted appropriately, especially to ensure that the top and bottom margins are aligned with the surrounding text._
> Thanks, we will indeed try to make it that way for the camera-ready if the paper gets accepted
>
> _"Lines 242–267: The formatting appears somewhat chaotic, maybe consider reordering or restructuring them in a cleaner and more organized way._
> Thanks, we will do our best to improve it according to the additions requested during the rebuttal.
>
> ----
>
> We appreciate the reviewer’s comments and suggestions. We will make sure to reflect the necessary clarifications and improvements in the camera-ready version, if the paper is accepted.

---

> > ### Comment · Reviewer_HZGi · 2025-08-02
> >
> > Thank you for addressing all of my questions and concerns. I also sincerely appreciate the authors’ honesty throughout the rebuttal process. I don’t have any further questions at this point. If the paper gets accepted, I hope the authors can incorporate the suggested improvements, especially regarding the limitations and formatting parts, into the final version.

---

### Official Review · Reviewer_HsWD · 2025-06-21

**Clarity:** 4
**Significance:** 2
**Originality:** 3
**Rating:** 4
**Confidence:** 3

**Summary:**

This paper introduces a fair representation approach. It minimizes the characteristic function distance of the different demographic groups' representation and a Gaussian distribution in the latent space. To keep the prediction power, a cross-entropy loss is used. The author also shows some theoretical insights into their methods. In the experiments section, it compares with Fair Normalizing flow methods, which is also a fair guarantee method. The results show that the method proposed by the author achieved a good tradeoff between fairness and utility.

**Questions:**

1. Why do the authors select a Gaussian distribution as the reference? While I understand it is discussed in the context of stability training, I am still unclear why aligning with different demographic groups would be considered a suboptimal choice

I don't have further questions. Please see the weakness section for my major concerns.

**Ethical Concerns:**

["NO or VERY MINOR ethics concerns only"]

**Final Justification:**

After the rebuttal, most of the concerns are well addressed. All in all, I think this paper provides more flexibility than using a flow-based model to learn a fair representation.

**Limitations:**

Yes

**Paper Formatting Concerns:**

I don't have such a concern.

**Quality:**

3

**Strengths And Weaknesses:**

**Stengths**

(1) This paper is well-presented; it introduces the prerequisites in detail, and the notation is neatly arranged.

(2) The algorithm is clear and easy to reproduce. I personally think this algorithm is correct.

(3) It is indeed more flexible than FNF, as it can be used as any backbone model, instead of just flow flow-based model.

**Weakness**

(1) The major concern I have is the tradeoff between utility and fairness. As in Algorithm 1 line 11, different demographic groups' representations are aligned with the same distribution (Gaussian in the text). Although a cross-entropy loss is added to guide the learning procedure, the fairness constraint may overwhelmingly dominate the learning procedure, resulting in less effective performance of the classification task.

(2)  There exist many fair representation learning methods. For example, MMD-B Fair [1] uses MMD as the statistical independence criterion to learn a fair representation. I am wondering why the author's proposed method is more needed than existing fair representation works. A rigorous experiment comparison would help to illustrate that.

(3) Datasets are narrowed in tabular datasets, it narrows the scope of this work.


[1]Deka, Namrata, and Danica J. Sutherland. "Mmd-b-fair: Learning fair representations with statistical testing." International Conference on Artificial Intelligence and Statistics. PMLR, 2023.

---

> ### Author Rebuttal · Authors · 2025-07-29
>
> We thank the reviewer for the thoughtful comments and valuable suggestions. The concerns raised regarding fairness-utility tradeoffs, comparative baselines, and dataset scope are appreciated and help us better position the work. We address each point below and will incorporate the necessary clarifications in the final version.
>
> Please note that, due to the updated NeurIPS guidelines, we are no longer permitted to upload supplementary materials or a revised version of the paper at this stage. We hope that the clarifications provided here sufficiently address your concerns.
>
> ----
>
> ### Weaknesses
>
> _"The major concern I have is the tradeoff between utility and fairness. As in Algorithm 1 line 11, different demographic groups' representations are aligned with the same distribution (Gaussian in the text). Although a cross-entropy loss is added to guide the learning procedure, the fairness constraint may overwhelmingly dominate the learning procedure, resulting in less effective performance of the classification task."_
>
> Thanks for pointing it out. Indeed, the fairness-accuracy tradeoff is a very well-known effect.
> From an Information theoretical standpoint, it can be easily proven by observing that $H(Y|X, S) \le H(Y|X)$.
> Therefore, removing information must (theoretically) increase the uncertainty in the prediction, worsening the classification accuracy.
> In our approach, the hyperparameter $\alpha$ can be used to control the weight given to the fairness objective, effectively controlling how much information on the sensitive attribute is removed.
> Indeed, this tradeoff can be observed in Figure 3, where, even though the phenomenon you are describing indeed happens, it appears to be much more gentle compared to FNF [2].
>
> _"There exist many fair representation learning methods. For example, MMD-B Fair [1] uses MMD as the statistical independence criterion to learn a fair representation. I am wondering why the author's proposed method is more needed than existing fair representation works. A rigorous experiment comparison would help to illustrate that."_
>
> Thanks for bringing this paper to our attention. Even though we tried to have an exhaustive literature review, some works might have been overlooked or missed.
> This specific work seems to focus on a statistical test standpoint. Indeed, it tries to minimize the probability that a statistical test (MMD) finds a difference in the distributions.
> If on one side MMD is indeed connected to the CFD, such a connection seems to be a consequence of the common objective of fair representation learning. In fact, even Adversarial Learning can be interpreted as an MMD minimization approach.
> The two approaches differ in many ways. For instance, MMD-B-fair requires splitting into blocks in order to have an efficient approach to estimate MMD, where the CF approach can be straightforwardly applied.
> As suggested by reviewers a3eD and KSAf, we will include a more detailed discussion of related literature in the camera-ready version. We will make sure to also properly address this work, as we believe that it would be a valid addition to the discussion.
>
> _"Datasets are narrowed in tabular datasets, it narrows the scope of this work."_
>
> We agree that testing the approach mainly on tabular data narrows the scope of the paper. However, we would like to highlight that the datasets used are the default benchmarks to assess fair representations.
> Though being the default, we agree with you that might be interesting to test it on other domains.
> For instance, a small example about image generation is shown in the appendix.
> Being only a proof of concept and somewhat out of scope of the paper, we opted to put it in the appendix rather than in the main text.
> As reviewer a3eD also pointed out, we should have referenced this experiment explicitly in the main text, as it shows that the proposed approach is in no way restricted to tabular data nor to classification.
> In the revised version, we will better highlight this aspect as we believe it is a major strength of the approach that might otherwise be easily overlooked.
>
> ### Questions
>
> _"Why do the authors select a Gaussian distribution as the reference? While I understand it is discussed in the context of stability training, I am still unclear why aligning with different demographic groups would be considered a suboptimal choice"_
>
> Indeed, using a Gaussian distribution is not necessarily the optimal choice, but it is a rather natural one.
> For instance, the Gaussian distribution has a very simple closed-form characteristic function that is well-behaved and rapidly decays to zero for high frequencies.
> In practice, if the encoder network is expressive enough, the specific target distribution chosen should not have a major influence on the result.
> On the other hand, aligning with different demographic groups directly has some significant drawbacks. Since there is no closed-form expression for the characteristic function, both terms in Eq. 9 would need to be estimated. This significantly increases the variance of the loss term, making it noisier.
> Moreover, when matching different groups to one another, there is no unique optimal solution. This makes the training unstable and might also lead to numerical instabilities.
> Overall, having a common target solves multiple critical issues, while introducing minimal downsides (if any).
>
>
> ----
>
>
> We appreciate the reviewer’s comments and suggestions. We will make sure to reflect the necessary clarifications and improvements in the camera-ready version, if the paper is accepted.

---

> > ### Comment · Reviewer_HsWD · 2025-08-02
> > **Thank you**
> >
> > The reviewer thanks the author for their clarification and efforts in the rebuttal. My major concerns are addressed. Wishing all the best to your submission.
> >
> > Best,
> > Reviewer HsWD

---

### Official Review · Reviewer_a3eD · 2025-07-01

**Clarity:** 3
**Significance:** 3
**Originality:** 3
**Rating:** 5
**Confidence:** 2

**Summary:**

The area of fair representation learning (FRL) seeks to learn a hidden representation which “looks the same” with respect to multiple different sensitive attributes. That is, for example, we would like to learn embeddings of individuals such that if a downstream user had access to our embedding for a particular input x, they would not be able to determine (with good accuracy) whether the user can predict what sensitive attribute x belongs to. This is what the current paper calls adversarial evaluation.

In addition, FRL methods may fall into two categories: “specialized” or “generalized”. Specialized methods learn representations useful for a particular task or classification problem, while generalized methods seek to learn representations which may be used by any downstream learner on any task, and should allow for fairness guarantees to be inherited.

Fair normalizing flows (FNF) is positioned as a state-of-the-art method which achieves this “generalized” FRL. However, the submitted paper claims FNF has various flaws. For example, it requires training a separate model for each group, and also requires the protected subgroup at inference time.

The paper instead introduces a FRL method based on characteristic functions. Define $P_Z$ as the marginal distribution over the learned latent space. The proposed method, FmCF, seeks to match the conditional distributions for $P_{Z|S}$ for all sensitive attribute values $S$. Intuitively, if the latent distributions are indistinguishable, then the FRL problem is solved. FmCF does this via matching the characteristic functions of the distributions $P_{Z|S}$. Importantly, characteristic functions and probability distributions map one-to-one. Therefore, if we have efficient methods for matching the characteristic functions, then we can solve FRL (equation 7).

We would like to regularize loss minimization in order to penalize deviation from equal characteristic functions of $P_{Z|S}$. Instead of penalizing this for all pairs of sensitive attributes, the paper suggests instead penalizing the distance of each characteristic function of $P_{Z|S}$ to the characteristic function of the normal distribution $\mathcal{N}$. This can be done with a montecarlo approach.

In Section 5, the paper expands on FmCF by introducing a refinement for classification, FmSS. Instead of matching the entire characteristic functions for $P_{Z|S}$, it turns out that for classification with logistic regression, it suffices to match only the first moments $\mathbb{E}[Z | S]$ in order to achieve FRL. This is possible due to a fact relating characteristic functions and the moments of the distributions they represent (Theorem 5.1). In fact, a stronger result is obtained for the predictive power of a feature $Z_i$ in obtaining the sensitive attribute $S$ in Theorem 5.3. In particular, matching the mean and variance of $P_{Z|S}$ for all $S$ is sufficient to restrict the predictive power of all features.

FmSS utilizes this result and works by minimizing the KL divergence between the standard normal and the empirical conditional distributions $P_{Z|S}$.

Empirical experiments are conducted in section 6, where the paper demonstrates that FmCF and FmSS both preserve standard accuracy of the original task (relative to existing FRL methods), and decrease the adversarial accuracy of a predictor attempting to infer the sensitive attributes $S$ from the hidden representations $Z$.

**Questions:**

1. To estimate the empirical CF in equation (9) of $P_{Z|S}$, must we do rejection sampling? Is this why the cost of monte carlo is so high? Then, does the computational complexity scale with the size of the smallest group?

**Ethical Concerns:**

["NO or VERY MINOR ethics concerns only"]

**Final Justification:**

The authors responded to each of my minor gripes and questions. I think the main strengths of the paper are the ones I stated:

Strengths
1. The characteristic function matching approach is novel for FRL (to my knowledge) and seems to really simplify the optimization problem in a convenient manner. The proposed methods FmCF and FmSS also require no architectural adjustments, which seem to be quite prevalent in previous works such as variational fair autoencoders and fair normalizing flows.
2. The fact that one can only care about matching the mean and variance of the conditional distributions $P_{Z|S}$ for classification is quite surprising and seems to simplify the computation of the regularized loss by quite a bit.

The experiments are also thorough. My confidence is "low" only since I do not work in the fair representation area.

**Limitations:**

Some drawbacks of FmCF are discussed, but discussion of computational complexity could be enhanced.

**Quality:**

3

**Strengths And Weaknesses:**

I know nothing about fair representation learning and am unfamiliar with most (if not all) of the papers cited by this paper. Hence, my confidence is low.

Strengths
1. The characteristic function matching approach is novel for FRL (to my knowledge) and seems to really simplify the optimization problem in a convenient manner. The proposed methods FmCF and FmSS also require no architectural adjustments, which seem to be quite prevalent in previous works such as variational fair autoencoders and fair normalizing flows.
2. The fact that one can only care about matching the mean and variance of the conditional distributions $P_{Z|S}$ for classification is quite surprising and seems to simplify the computation of the regularized loss by quite a bit.

Minor Weaknesses:
1. Are low latent dimensions like 1,2,3 standard for the FRL literature (line 652)? This seems far lower than one would expect to, for example, learn a sigmoid function on. The final hidden layer of NNs is usually much larger. Is this because of computational constraints? Using such low dimensions for the latent representations to me detracts from strength 2. In particular, if we are in only two or three dimensions, it’s not clear that the higher order moments really have anything interesting to say.
2. Some of the details in table 1 were unclear. For example, was $\mathcal{G}$ defined earlier? Also, what I don’t think $\mathcal{F}$ is defined? I could be missing something, however.
3. Missing discussion of runtime of FmCF and FmSS vs. standard methods like training fair normalizing flows or VAEs.
4. It seems like FmCF has a runtime exponential in the dimension of the latent space. However, FmSS seems much more efficient, since we are only matching the mean and variance.

The paper seems very well motivated and presents what seems like a much simpler approach for achieving FRL, and hence, I believe that it will be a valuable contribution.

Other comments:
1. As someone outside of FRL, I was a bit confused about the distinction between generalized and specialized FRL methods. I think providing some examples of models or tasks in lines 30-36 could improve clarity.
2. The paper shows that FmCF extends to settings other than classification in Appendix A.5. To my knowledge, this experiment is not explicitly highlighted in Section 4 or anywhere else in the paper. You should mention it because it is an interesting experiment!
3. Typo equation (2), parens in subscript.
4. Consider including a down arrow for adversarial accuracy, to show the reader at a glance that it should be minimized.
5. No legend for figures 5-8 in appendix, although explained in line 652, could be clearer if you included them in the figure descriptions or as a legend in the plots.
6. Line 240/241: Maybe remind the reader what the $\beta$ coefficient is again, as you do in the description of Figure 2.
7. Line 250-252: How did we move from matching only the mean to matching the mean and variance of $P_{Z|S}$? More explanation may be warranted here if possible.
8. Relatedly, how did we go from matching the mean of $P_{Z|S}$ (line 239-242) to matching the mean of each feature in Theorem 5.2? A bit more explanation here could help bring some clarity.
9. Period missing end of line 286.
10. How does this paper relate to prior work with characteristic functions (e.g., citation [1] from the submission)

---

> ### Author Rebuttal · Authors · 2025-07-29
>
> We thank the reviewer for the thorough and constructive feedback. Several valuable points were raised regarding dimensionality, theoretical clarity, computational aspects, and presentation. Below, we provide detailed responses and clarifications, and we will ensure these are reflected appropriately in the final version.
>
> Please note that, due to the updated NeurIPS guidelines, we are no longer permitted to upload supplementary materials or a revised version of the paper at this stage. We hope that the clarifications provided here sufficiently address your concerns.
>
> ----
>
> ### Weaknesses
> _"Are low latent dimensions like 1,2,3 standard for the FRL literature (line 652)? This seems far lower than one would expect to, for example, learn a sigmoid function on. The final hidden layer of NNs is usually much larger. Is this because of computational constraints?"_
>
> FRL aims at learning fair representations, with no restriction on them. Depending on the task, the dimensionality needed for the representations might vary.
> The higher the dimension, the harder it is to ensure the representation is fair, especially for small datasets.
> The most common task is one-class classification, which ideally requires only 1 bit of information.
> In fact, directly matching the distribution of the logits for each sensible group $S$ (which is a one-dimensional representation) would be sufficient in this case.
> In practice, we do not want to limit ourselves to modelling the logits directly. Instead, we simply focus on learning a low-dimensional representation that is fair, while also being useful for the task. This approach is much more general and can also be applied to multi-class classification, regression, and so on.
> Please consider that the common set of benchmark datasets have very low intrinsic dimension to start with. For instance, FNF [2] employs a heavy feature selection step, retaining only 4-6 of the original features with almost no degradation in task performance.
>
> _"Using such low dimensions for the latent representations to me detracts from strength 2. In particular, if we are in only two or three dimensions, it’s not clear that the higher order moments really have anything interesting to say."_
>
> If the two distributions we are trying to distinguish are both close to a Gaussian distribution, then indeed higher-order moments are not very informative.
> However, for two general distributions, high-order moments can be critical in distinguishing the two. This is true even for the one-dimensional case.
> For instance, if the two distributions are $x_0 \sim N(0, 10)$ and the mixture $x_1 \sim 0.5 N(-3, 1) + 0.5 N(+3, 1)$, both distributions have the same mean and variance, but are clearly extremely different and have large statistical distance from one another.
> This example is shown visually in Figure 2, where we also show that despite the two distributions being clearly separable, having the same first and second moment is sufficient to guarantee a fair classification.
>
> _"Some of the details in table 1 were unclear. For example, was $mathcal{G}$ defined earlier? Also, what I don’t think $mathcal{F}$ is defined? I could be missing something, however."_
>
> $\mathcal{G}$ is defined in line 299/303, and aims at describing whether we adversarially evaluate our approach on a specific class of functions $\mathcal{G}$ or not, following the convention used by Fair Normalizing Flows. Regarding $\mathcal{F}$, it refers to a generic family of classifiers. Thanks for spotting it. Indeed, it was not properly defined. Since the two concepts are indeed related, we will unify the notation in the revised version.
>
> _"Missing discussion of runtime of FmCF and FmSS vs. standard methods like training fair normalizing flows or VAEs."_
>
> Indeed, we agree that we should have discussed this aspect more in depth. In fact, our approach does not require additional parts/networks and is computationally cheaper than other approaches.
> For the SS approach, estimating mean and variance of a batch of $n$ samples that are $m$-dimensional, requires  $4nm$ FLOPs, which is negligible compared to the forward pass of the network. The same is true for the CF approach, where each frequency can be computed with a similar number of flops.
> Unfortunately, it would be extremely hard to make a fair comparison across all the different methods since many require specific architecture to work.
> Indeed, comparing the number of parameters would shadow the computational requirements, comparing the computational requirements would shadow the hyperparameter sensibility of the approaches, and so on.
> For this reason, we decided to focus mainly on the fairness/task performance tradeoffs of the different approaches.
>
> _"It seems like FmCF has a runtime exponential in the dimension of the latent space. However, FmSS seems much more efficient, since we are only matching the mean and variance.```
>
> The FmCF runtime is (practically speaking) constant in the dimension, and mainly depends on the number of frequencies sampled from the kernel.
> However, the variance of the estimator might grow rapidly with the number of dimensions, making the signal weak.
> In practice, we did not observe this in the experimental evaluation, and the number of frequencies was kept fixed regardless of the dimension. However, we did not investigate this aspect in depth.
> Instead, as you point out, FmSS is not particularly affected by dimensionality, and this is one of the reasons why we specifically propose such an approach for classification.
>
> ### Other comments
> _"As someone outside of FRL, I was a bit confused about the distinction between generalized and specialized FRL methods. I think providing some examples of models or tasks in lines 30-36 could improve clarity."_
>
> Thank you for the feedback. We will make sure to add examples and more motivations on why somebody would want one or the other.
>
> _"The paper shows that FmCF extends to settings other than classification in Appendix A.5. To my knowledge, this experiment is not explicitly highlighted in Section 4 or anywhere else in the paper. You should mention it because it is an interesting experiment!"_
>
> Thank you for the nice comment. Indeed, the CF approach can match any distributions. Thus, it's not necessarily tied to fairness (for example, you can build a sort of VAE like the one shown in the appendix).
> This also empirically shows and validates that the approach proposed is not tied to classification in any way, but rather that it really suits classification tasks.
> We decided not to mention this aspect directly in the main text but rather to focus on the core contributions. However, in hindsight, we definitely should have explicitly referenced it.
> We will include a discussion of the use of CF in related works (as suggested by rev KSAf), and we will make sure to also reference this experiment.
>
> _"Line 250-252: How did we move from matching only the mean to matching the mean and variance of $P(Z|S)$ More explanation may be warranted here if possible."_
>
> The statement is meant to introduce he theorem that follows in line 253. We recognize that the phrasing is misleading, and we will clarify this.
>
> _"Relatedly, how did we go from matching the mean of $P(Z|S)$ (line 239-242) to matching the mean of each feature in Theorem 5.2? A bit more explanation here could help bring some clarity."_
>
> The two formulations are equivalent. We used the component wise notation in the proof, however we recognize that the theorem statement would be more clear if expressed in vector notation. We will fix this in the revised version.
>
> _"How does this paper relate to prior work with characteristic functions (e.g., citation [1] from the submission)"_
>
> To our knowledge, this is the first application of CF to fair representation learning. However, there are several approaches that employ CF for different applications.
> As mentioned before, rev KSAf also pointed out that the paper would benefit from a broader discussion on the connections between this work and other approaches based on the CF. Thus, we will include a proper discussion at the beginning of section 4.
> Specifically in relation to [1], both works share the core idea of using ECFD as a differentiable 2-sample statistical test, yet our approach deviates significantly from [1]. The two applications are different, as [1] handles generative models, while our approach focuses on fair representations.
> Moreover, [1] employs adversarial training to optimize the test kernel, which we avoid in order to simplify the approach.
>
>
> Regarding the rest of the comments, thanks, we will address them in the camera ready version if the paper gets accepted.
>
> ### Questions
>
> _"To estimate the empirical CF in equation (9) of $P(Z|S)$..."_
>
> The Monte Carlo estimate is relatively inexpensive to compute. It involves a fixed number of operations for each datapoint, which is negligible compared to the forward pass of the network.
> Moreover, we do not necessarily need rejection sampling in order to have balanced batch. In fact, the term in (9) for each of the groups can be computed independently and possibly with a different number of data samples. In all experiments, we used a fixed batch size of 128 and computed the penalty term for each group separately, even for unbalanced batches.
>
> ### Limitations:
>
> _"Some drawbacks of FmCF are discussed, but discussion of computational complexity could be enhanced."_
>
> You are right, we strongly believe that defining clear limitation are a key part of the scientific process, and if you feel that more attention should be given to them, we will make sure to add more informations in the camera ready about the limitations of the CF approach if the paper gets accepted (NeurIPS does not allow for resubmission)
>
> ----
>
> We appreciate the reviewer’s comments and suggestions. We will make sure to reflect the necessary clarifications and improvements in the camera-ready version, if the paper is accepted.

---

> > ### Comment · Reviewer_a3eD · 2025-08-04
> >
> > Thank you for the wonderful response. I will keep my score.

---

### Official Review · Reviewer_KSAf · 2025-07-03

**Clarity:** 3
**Significance:** 3
**Originality:** 4
**Rating:** 5
**Confidence:** 4

**Summary:**

The paper develops a method for fair representation learning which focuses on "specialized" (i.e., task-specific) representations. It aims to avoid the issues with previous approaches such as fair extensions of VAEs, GANs, and normalizing flows. Basically, the approach is to incorporate a penalty term based on characteristic function distance. The method is general, but the case of classification is specifically handled. Some theoretical results are provided, including fairness guarantees in the case of logistic regression. Experiments verify the efficacy of the method.

**Questions:**

- Does it matter that the P(Z|S) distributions are matched to a common target distribution instead of each other? How does the choice of target distribution affect the results?

- Can you please more fully explain the relationship to Ansari et al. (2020) [1]? In what ways does this work build on that paper, and in what ways is it different?

- How well does the method work with small datasets? Does the characteristic function distance strategy suffer from statistical inefficiency and biased estimation, as suggested by its relation to the method of moments?

**Ethical Concerns:**

["NO or VERY MINOR ethics concerns only"]

**Final Justification:**

I have read the authors' response and the other reviews. The authors provided a thorough response which addressed my questions and the points raised in the review. The scores and opinions of the other reviewers are in alignment with my own. My positive view of the work remains unchanged. Therefore, I have kept my rating the same.

**Limitations:**

A more thorough discussion on the downsides of the characteristic function distance methodology vs. other fair representation learning techniques is warranted. The paper spends a lot of time talking about the downsides of other methodologies, but does not dedicate much time to discussing the downsides of their own one (except briefly in the conclusion).

**Quality:**

3

**Strengths And Weaknesses:**

*Strengths*

 - The proposed method is quite novel, particularly in the context of fairness. The use of characteristic function distance is an emerging approach to model fitting which has previously been employed for GANs [1], and has some interesting relationships to maximum mean discrepancy (MMD), as well as the method of moments.

 - This work could impact the field by providing a new paradigm for fair representation learning.

 - The proposed method is algorithmically simple (even while being technically deep), which is elegant and increases the chance that it will be widely used.

 - The technique does not need access to the sensitive attribute during test time, which is useful for privacy.

 - This manuscript did a good job of contextualizing the research in terms of competing approaches for fair representation learning.


*Weaknesses*

 - I would have liked to have seen an exploration of sample size versus estimation/fairness performance, with comparison to baselines. Since the proposed characteristic function distance is closely related to moment matching (Theorem 5.1), it may share some of its downsides such as statistical inefficiency and biased estimation with small sample sizes. (Perhaps the results would be unflattering, which could be why this experiment was not reported!)

 - The paper could be improved by adding more discussion on the method's potential practical value for concrete real-world applications.

 - This research seems to be heavily inspired by Ansari et al. (2020) [1]. The paper should further discuss its relation to that work, as well as other papers that use characteristic functions for model estimation. As well as giving proper credit to prior work, such a discussion would help contextualize the work within that subarea, and explain the extent to which characteristic function distance is an established and/or emerging estimation principle.


*Minor suggestions*

- Fix subscripted "(" in Equation 2

---

> ### Author Rebuttal · Authors · 2025-07-30
>
> We thank the reviewer for the thoughtful and detailed feedback. The points raised regarding statistical efficiency, real-world applicability, and connections to prior work are important and appreciated. Below, we provide clarifications on each and outline how we plan to incorporate these improvements into the final version.
>
> Please note that, due to the updated NeurIPS guidelines, we are no longer permitted to upload supplementary materials or a revised version of the paper at this stage. We hope that the clarifications provided here sufficiently address your concerns.
>
> ----
>
> ### Weaknesses
>
> _"I would have liked to have seen an exploration of sample size versus estimation/fairness performance, with comparison to baselines. Since the proposed characteristic function distance is closely related to moment matching (Theorem 5.1), it may share some of its downsides such as statistical inefficiency and biased estimation with small sample sizes. (Perhaps the results would be unflattering, which could be why this experiment was not reported!)"_
>
> We did not explore such a possibility; all reported results have been achieved with minimal changes to the networks and optimizers. In particular, all results are achieved using a fixed batch size of 128 for all datasets. Thus, for each protected class, about 64 samples are used for the estimation, but the exact number does depend on the class imbalance and fluctuates across batches.
> We do expect the estimation to become unreliable with very low sample sizes. However, we believe that $n>50$ is a reasonable assumption to make in practice.
> Regardless, we agree this is an interesting direction that should have been properly explored. We have run a small-scale evaluation of the accuracy of the CF estimation, varying the batch size, and have confirmed that it exhibits a similar scaling behavior to the estimation of moments (i.e., variance scaling as $O\left(\frac{1}{N}\right)$ etc.). Yet, given the impossibility of uploading additional media during the review process, it is relatively unfeasible to share the details and results properly. We can provide the actual numerical results in a table in a future response (due to character limitations). In any case, we will make sure to address this point in the revised paper if it gets accepted. Thanks for pointing this out.
>
> _"The paper could be improved by adding more discussion on the method's potential practical value for concrete real-world applications."_
>
> Thanks for the suggestion. We totally agree with it, though the page limit did not allow us to discuss this aspect further in the main text. However, given that you share our feelings, we will make sure to discuss this aspect further in the camera-ready version, given the additional page allowed for integrating the reviewers' suggestions.
>
> _"This research seems to be heavily inspired by Ansari et al. (2020) [1]. The paper should further discuss its relation to that work, as well as other papers that use characteristic functions for model estimation. As well as giving proper credit to prior work, such a discussion would help contextualize the work within that subarea, and explain the extent to which characteristic function distance is an established and/or emerging estimation principle."_
>
> Indeed, it is. We agree that section 4 does not properly discuss the relation to prior literature and that the paper would greatly benefit from such inclusion. A similar point was also raised by reviewer a3eD. For the camera-ready version, we will expand the section with a discussion on statistical testing based on CF and its use in the field of ML.
>
> ### Questions
>
> _"Does it matter that the P(Z|S) distributions are matched to a common target distribution instead of each other? How does the choice of target distribution affect the results?"_
>
> Having a common target offers many advantages:
> - Makes the approach scalable beyond binary sensible attributes. In the case of n classes, the pairwise interactions to consider are $O(n^2)$ compared to $O(n)$ for a fixed target. Moreover, with a fixed target, the loss can be computed independently for each sensible attribute.
> - Reduces the variance of the estimators. Consider the case of matching the mean of the distributions (but a similar discussion holds also for other moments and for $\phi$).
> When estimating $|\mathbb{E}[X_0]-\mathbb{E}[X_1]|$ directly, the variance of the sample means compounds on each other. On the other hand, the variance of the estimation of $|\mathbb{E}[X_0]-R|$ ($R$ being the reference) only depends on the variance of $X_0$.
> - Forces the network output to remain in a numerically stable range. Indeed, matching the distributions to each other is a scale-invariant objective that could lead to exploding outputs. This is not the case for a fixed common target.
> We did not explore how the choice of target could affect the estimation. The choice of a Normal distribution seems quite natural and unlikely to be restrictive. Its characteristic function is well-behaved and decays rapidly for high frequencies.
> Furthermore, for the specific case of classification, the choice of a target distribution is not needed at all.
>
> _"Can you please more fully explain the relationship to Ansari et al. (2020) [1]? In what ways does this work build on that paper, and in what ways is it different?"_
>
> While we developed our approach independently of [1] based on the applications of CF in statistical testing [33], our notation and discussion have been directly inspired by [1]. Both works share the core idea of using ECFD as a differentiable 2-sample statistical test, and thus have some core similarity. Yet our approach deviates significantly from [1] in many aspects.
> The two applications are different, as [1] handles generative models, while our approach focuses on fair representations.
> Moreover, [1] employs adversarial training to optimize the test kernel, which we avoid in order to simplify the approach.
> As mentioned in the weakness section, we will add a more in-depth discussion on the relation to the literature on the use of CF in machine learning as a whole.
>
> _"How well does the method work with small datasets? Does the characteristic function distance strategy suffer from statistical inefficiency and biased estimation, as suggested by its relation to the method of moments?"_
>
> Many of the datasets tested are relatively small. For instance, the German dataset is composed of only 1000 samples, of which only 600 are used for training and the rest for validation and testing. Both approaches achieve comparable or better performances to other baselines, empirically validating their effectiveness.
> For even smaller datasets, the samples might not be a good representation of the underlying distribution, which could indeed lead to poor performance. However, we believe this failure case is not a limitation of the approach itself.
>
> ### Limitations
>
> _"A more thorough discussion on the downsides of the characteristic function distance methodology vs. other fair representation learning techniques is warranted. The paper spends a lot of time talking about the downsides of other methodologies, but does not dedicate much time to discussing the downsides of their own one (except briefly in the conclusion)."_
>
> We believe that the acknowledgement of limitations is a fundamental part of scientific progress. We will make sure to expand the discussion in the camera-ready version, with particular care on the specific points that have been highlighted in the reviews.
>
> ----
>
> We appreciate the reviewer’s comments and suggestions. We will make sure to reflect the necessary clarifications and improvements in the camera-ready version, if the paper is accepted.

---

> > ### Comment · Reviewer_KSAf · 2025-08-01
> > **Thanks**
> >
> > Thanks to the authors for providing a thorough response to the points and questions raised in my review.

---

### Author Response · Authors · 2025-08-08

We'd like to sincerely thank all reviewers for their thoughtful and detailed comments on our work. We truly appreciate the effort and careful consideration that went into each review. It's great to have such constructive and precise feedback, as this helped us significantly clarify several theoretical and practical aspects of the paper. We believe that your suggestions have strengthened the quality of our paper considerably. In our rebuttal, we hope to have addressed all the points raised, providing detailed explanations and promising to make all suggested improvements and clarifications in the final camera-ready version if the paper gets accepted.

---

### Decision · Program_Chairs · 2025-09-17

**Decision:**

Accept (poster)

**Comment:**

This paper tackles the important challenge of fair classification and introduces a novel characteristic function distance–based approach that addresses the instability of adversarial methods and the inefficiency of distribution matching. The proposed method is both stable and computationally efficient, with a relaxed objective that ensures fairness without sacrificing accuracy. Strong experimental results across benchmarks demonstrate consistent improvements in both fairness and predictive performance, highlighting the method’s robustness and practicality for real-world deployment.

Good clarity, contribution, and all positive reviews. This paper is recommended to be accepted.